# APOBEC3 promotes squamous differentiation via IL-1A/AP-1 signaling

Michael S. Sturdivant ●[1,2,7], Andrew S. Truong[1,2,7], Mi Zhou ●[1], Elliott D. Toomer[1,2], Wolfgang Beckabir[1], John Raupp[1], Ujjawal Manocha[1], Ibardo A. Zambrano[3], Hung-Jui Tan[3], Marc A. Bjurlin[3], Angela B. Smith[3], Tracy L. Rose[1,4], Matthew I. Milowsky ●[1,4], Sara E. Wobker[1,5], Kathryn H. Gessner[1,3,6], Jeffrey S. Damrauer ●[1,4,8] ✉ & William Y. Kim ●[1,2,4,6,8] ✉

The APOBEC3 family of RNA and single stranded DNA cytidine deaminases contribute prominently to the mutagenesis of certain cancers including urothelial carcinoma of the bladder (UC). Remarkably, up to 70% of mutations in UC are attributable to the mutagenic activity of the APOBEC3 deaminases. Despite this strong association, few functional studies have investigated APOBEC3's role in bladder cancer. We report a genetically engineered murine model with conditional knock out of *Pten* and *Trp53* in addition to overexpression of mouse Apobec3 (UPPA). Analysis of bladder tumors from UPPA mice demonstrates that mA3 promotes tumor progression and squamous trans-differentiation. We establish that APOBEC3 promotes squamous differentiation through IL-1α and downstream activation of the AP-1 transcription factor. Bulk RNA-sequencing from human UC shows *APOBEC3A* as the only human APOBEC3 family member to correlate with squamous differentiation. Furthermore, single cell and spatial transcriptomics reinforces the role of *APOBEC3A* in fostering squamous trans-differentiation and promoting the emergence of a subpopulation of highly squamous epithelial cells. Our results demonstrate that mouse Apobec3 and human APOBEC3A promote squamous differentiation in urothelial carcinoma and that this trans-differentiation phenotype is mediated through IL-1α signaling, a target of FDA approved therapies for rheumatologic disease.

With ~85,000 new cases and 17,000 deaths estimated in 2025, bladder cancer (BCa) is highly prevalent in the United States, affecting 1 in 29 men during their lifetime[1]. BCa can be histologically subtyped, with urothelial carcinoma (UC) (i.e. conventional) being the most common. While conventional UC represents the majority of bladder cancer diagnoses, there is a range of histologic subtypes (micropapillary, plasmacytoid, sarcomatoid) as well as UC with divergent differentiation, including squamous, glandular, and trophoblastic features[2,3]. Individuals may exhibit a combination of conventional UC, subtypes, histology and divergence within the same tumor. Clinically, BCa is

[1]Lineberger Comprehensive Cancer Center, University of North Carolina at Chapel Hill, Chapel Hill, NC, USA. [2]Department of Pharmacology, University of North Carolina at Chapel Hill, Chapel Hill, NC, USA. [3]Department of Urology, University of North Carolina at Chapel Hill, Chapel Hill, NC, USA. [4]Department of Medicine, Division of Oncology, University of North Carolina at Chapel Hill, Chapel Hill, NC, USA. [5]Department of Pathology, University of North Carolina at Chapel Hill, Chapel Hill, NC, USA. [6]Department of Genetics, University of North Carolina at Chapel Hill, Chapel Hill, NC, USA. [7]These authors contributed equally: Michael S. Sturdivant, Andrew S. Truong. [8]These authors jointly supervised this work: Jeffrey S. Damrauer, and William Y. Kim. ✉e-mail: jeffrey_damrauer@med.unc.edu; wykim@med.unc.edu

divided between non-muscle invasive (NMIBC) and muscle-invasive bladder cancer (MIBC). While NMIBC has a good overall prognosis (5-year survival rate 95%) as compared to muscle-invasive bladder cancer (MIBC) (5-year survival rate of 50%), NMIBC frequently recurs, can progress to MIBC, and ultimately become metastatic[1].

While conventional UC is the dominant histology, squamous differentiation occurs in up to 21% of cases and is a well-documented poor prognostic factor associated with higher rates of nodal metastases and local recurrence in both NMIBC as well as in MIBC[4]. Histologically, squamous regions within a UC tumor are defined by the presence of keratin pearls or intercellular bridges/desmosomes. Molecularly, squamous and non-squamous regions have similar recurrent somatic mutations; however, squamous regions often have increased karyotypic complexity and higher tumor mutational burden (TMB)[5]. Taken together, this suggests that while there may not be unique somatic mutations driving squamous differentiation, there could be unique biologic processes leading to increased levels of non-synonymous mutations and genome instability.

One way to understand oncogenic processes is through the analysis of synonymous and non-synonymous mutational patterns within the genome. These patterns of single-nucleotide variants (SNVs), referred to as mutational signatures, can be used to identify environmental mutagens and mutational processes that have contributed to tumor development. Over the past decade, it has been recognized that the most prevalent of these signatures within bladder cancer is a result of apolipoprotein B mRNA-editing enzyme, catalytic polypeptide (APOBEC) activity[6,7].

APOBEC is a superfamily of RNA and single-stranded DNA (ssDNA) cytidine deaminases that includes AID, APOBEC1, APOBEC2, APOBEC3, and APOBEC4 and their cognate family members[8]. In humans, the APOBEC3 family comprises seven members (APOBEC3A/B/C/D/F/G/H), while there is notably only a single Apobec3 family member in mice. In recent years, members of the APOBEC3 family, particularly APOBEC3A and APOBEC3B, have been studied as mutagenic drivers in cancer, although only a few studies have examined the functional role of APOBEC-induced mutagenesis[9–13]. As enzymes, APOBEC3A/B localize to the nucleus where[14,15] they catalyze the conversion of cytosine to uracil through deamination of cytosine in the context of a TCW (W = A or T) trinucleotide motif on ssDNA. Uracil-DNA glycosylases then remove the uracil residue via glycolytic cleavage between the base and the sugar backbone to create an abasic site. Specialized trans-lesion polymerases allow DNA synthesis to proceed over the DNA lesions, ultimately resulting in predominantly C-to-T and C-to-G mutations[9,16–18].

Recently, Liu et al. demonstrated that the transgenic overexpression of APOBEC3G can shorten survival and drive genomic instability in the N-butyl-N-(4-hydroxybutyl)nitrosamine (BBN) BCa mouse model[19]. Although BCa has one of the highest proportions of APOBEC-induced mutations, and *APOBEC3A* and *APOBEC3B* gene expression are significantly upregulated in tumor tissue relative to normal urothelium[12] the roles of *APOBEC3A* and *APOBEC3B* have not been systematically explored in bladder cancer. Additional evidence for the importance of APOBEC3 in bladder cancer is the association of germline risk variants in the APOBEC3 region that associate with bladder cancer risk[20].

To interrogate the role of APOBEC3 in UC initiation and tumor progression, we developed a Cre recombinase inducible Apobec3 (mA3) mouse allele, to allow for the temporal-spatial control of mA3 expression in the background of an autochthonous mouse model of bladder cancer[21]. We demonstrate that urothelial-specific mA3 expression promotes tumor progression and squamous differentiation in autochthonous bladder tumors. Furthermore, we identify the production of Interleukin-1 alpha (IL-1α) in response to mA3-induced DNA damage, and its subsequent downstream activation of the AP-1 transcription factor, as being sufficient to drive squamous-associated

gene expression. Finally, leveraging multi-omic and spatial transcriptomic data from primary tumors, we demonstrate that in humans, where multiple APOBEC3 subunits are expressed, *APOBEC3A* expression is most associated with histologic and molecular features of squamous differentiation.

## Results

### Apobec3 accelerates bladder tumor formation and promotes squamous differentiation in vivo

To address the role of APOBEC3 enzymes in bladder cancer, we knocked in the cDNA encoding mouse *Apobec3* (mA3) at the *Rosa26* locus under control of a CAG promoter and a LoxP-STOP-LoxP cassette (Supplementary Fig. 1A) (hereafter called *Rosa26^LSL-Apobec3*). The *Rosa26^LSL-Apobec3* mice were then crossed with our previously reported *UPP* (*Upk3a-Cre^ERT2; Pten^L/L; Trp53^L/L*) bladder cancer mouse model[21] to create *UPPA* mice. This *UPPA* (*Upk3a-Cre^ERT2; Pten^L/L; Trp53^L/L; Rosa26^LSL-Apobec3*) mouse model, conditionally overexpresses mA3, while simultaneously inactivating *Pten* and *Trp53* in Upk3a-expressing intermediate/umbrella cells (Fig. 1A). We chose to express mA3 rather than any of the human APOBEC3 family members to avoid the introduction of strong allo-antigens, which could confound any immunologic assays.

Tamoxifen was administered to both *UPP* (control) and *UPPA* mice every other day for three doses via oral gavage starting at 6–8 weeks of age to induce Cre^ERT2 activity. Animals were monitored by serial ultrasound imaging of the bladder (Fig. 1B). Despite no significant difference in bladder tumor incidence [UPP = 32/40 and UPPA = 37/40, Fisher's exact $p = 0.19$] or volume between *UPP* ($n = 6$) and *UPPA* ($n = 6$) mice (Supplementary Fig. 1B), *UPPA* mice had a significantly shorter tumor latency ($p < 0.001$), developing papillary bladder tumors at a median time of 43.3 weeks as compared to 53.6 months for *UPP* mice (Fig. 1C). Therefore, Apobec3 does not appear to play a role in the murine tumor initiation but rather increases tumor growth rate.

Multiple cell lines from both *UPP* ($n = 3$) and *UPPA* ($n = 3$) tumors were generated and evaluated for mA3 protein expression and deaminase activity to confirm the robustness of the Rosa26^LSL-Apobec3 allele. Endogenous levels of mA3 were expressed at similar levels across all the cell lines, with only *UPPA* cells expressing exogenous mA3 (Fig. 1D). Functionally, cytosine deaminase activity was elevated in all three *UPPA* cells as compared to *UPP* cell lines (Fig. 1E).

Next, hemotoxylin and eosin-stained slides of the murine tumors were reviewed in a blinded fashion by a fellowship trained genitourinary surgical pathologist to evaluate tumor histology and the presence of divergent differentiation. While both *UPP* and *UPPA* tumors predominately featured papillary histology (Fig. 1F), a significantly greater proportion of *UPPA* tumors exhibited squamous differentiation (Chi-square $p = 0.049$) (Fig. 1G), as noted by the presence of keratin pearls (Fig. 1F). When comparing the extent of squamous differentiation between UPP and UPPA tumors, we did not note a difference in the proportion of squamous histology per tumor (Supplementary Table 1).

### scRNA sequencing reveals UPPA-specific squamous subpopulations

We next wanted to understand what, if any, underlying gene expression patterns were specific to the APOBEC3-expressing cells within *UPPA* tumors. To this end, we performed scRNA sequencing on *UPP* ($n = 2$) and *UPPA* ($n = 2$) tumors to better guide our tumor characterization. RNA-seq reads were aligned to the *mm10* genome, as well as the 5'UTR/WPRE of the LSL-APOBEC3 allele to detect exogenous mA3 expression, and cells were clustered using Seurat[22]. Cell clusters were annotated for cell type based on the expression of canonical marker genes (Fig. 2A), and differential gene expression analysis was performed comparing the epithelial cell clusters from *UPPA* and *UPP* tumors. Aligning with the histologic findings, genes related to squamous differentiation, such as those involved in cornification of the

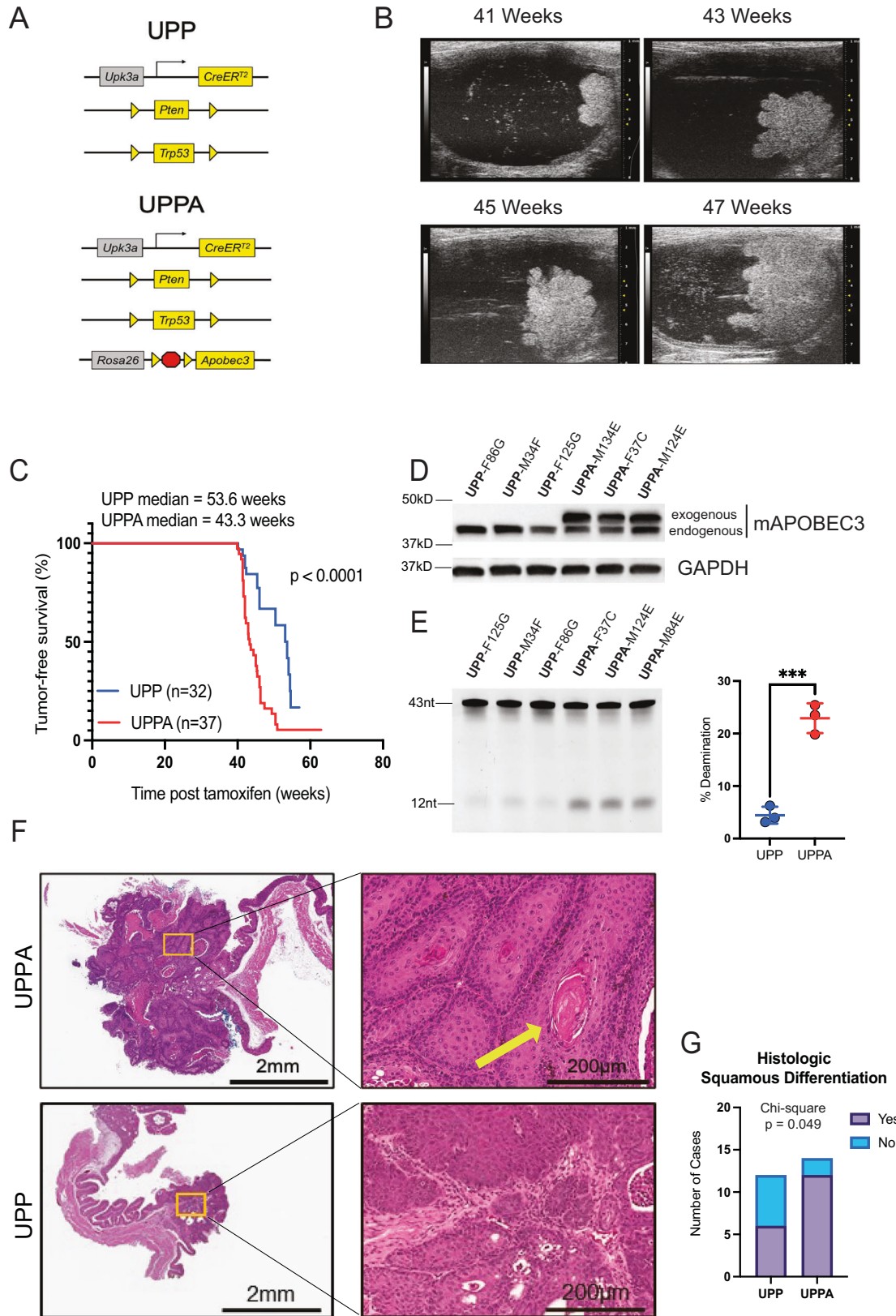

stratum corneum of skin (i.e., *Cnfn, Ivl, Sprr1a, Sprr1b, Sprr2b, Sprr2d*, etc.) and desmosome formation (i.e., *Dsg2, Dsg3*, etc.) were expressed at significantly increased levels in *UPPA* tumors as compared to *UPP* tumors ($p < 0.05$) (Fig. 2B).

To identify squamous subpopulations, we re-clustered only the epithelial cells from the prior analysis. Sub-clustering identified six

clusters of epithelial cells, which we have termed Basal 1, Basal 2, Basal 3, Intermediate 1, Intermediate 2, and Luminal, based upon their expression of marker genes for known urothelial cell types (Supplementary Fig. 2A, B). Pathway analysis of the top differentially expressed genes identified two clusters, Basal 1 and Intermediate 1, as enriched for genes related to the keratinization and development of a cornified

**Fig. 1 | Murine bladder tumors with Apobec3 overexpression have shortened tumor latency and a basal/squamous phenotype. A** Schematic showing the design of the UPP and UPPA mouse models. **B** Serial ultrasound images of a single UPPA mouse showing the growth of the bladder tumor. **C** Tumor-free survival curve comparing the time it took for bladder tumors to be detected by ultrasound imaging in a cohort of UPP and UPPA mice after tamoxifen treatment. Significance was calculated using the log-rank test, $p < 0.0001$. **D** Western blots of whole cell lysates from UPP (biologic replicates, $n = 3$) and UPPA (biologic replicates, $n = 3$) cell lines blotted with indicated antibodies. **E** Cytosine deaminase assay of whole cell lysates from independent UPP and UPPA tumor-derived cell lines. Quantification of lower band intensity between UPP and UPPA indicating percentage of deamination activity. Significance was calculated using a two-sided unpaired $t$-test. Data represent mean ± SD ($n = 3$, biological replicates). ***$p < 0.001$. **F** Representative H&E images of a UPP and a UPPA tumor at two different magnifications (2x and 20x) showing the presence of squamous differentiation. **G** Bar plot of UPP and UPPA bladder tumors showing the frequency of histologic squamous differentiation, determined by a fellowship trained GU pathologist. Significance was calculated using a $\chi^2$ test. Source data are provided as a Source Data file.

envelope, indicating a high level of squamous gene expression (Supplementary Fig. 2C). Due to their relatedness, we combined these two clusters (Basal 1 and Intermediate 1) and designated them as "squamous cells" in downstream analysis. The differentially expressed genes identified between the *UPPA* and *UPP* squamous cells were then compared against the *Descartes Fetal Squamous Epithelial* cell signature[23] to determine if known squamous genes were enriched in either of the squamous populations. While prior analysis classified both populations as squamous, 25 of the 36 genes that were expressed in both our datasets and the Descartes fetal squamous epithelial signature, had significantly higher expression ($p < 0.05$) in *UPPA* squamous cells as compared to *UPP* squamous cells (Fig. 2C).

Trajectory inference analysis was performed to better understand the intercellular relationships within the epithelial cells of UPPA and UPP tumors. Analysis comparing the epithelial cells indicated a differentiation pattern that progressed from node 0 to node 2 when plotted by Pseudotime (Fig. 2D). This differentiation pattern was also visualized when plotted by CellState, with increased differentiation moving from CellState 1 to CellState 4/5 (Fig. 2E). Not only did the distribution of cells among CellState between *UPP* and *UPPA* tumors differ, with *UPPA* tumor having increased proportions of cells belonging to CellState 4 and 5 (the more differentiated nodes), but the proportion of cells expressing and expression itself of Apobec3 (LSL and endogenous) increased from CellState 1–5. (Fig. 2F and Supplementary Fig. 2D–G). Based on the prior histologic and differential expression analysis, we postulated that CellState 4/5 cells were more squamous-like than the less differentiated cells in the analysis; this was confirmed by the progressive increase in mRNA expression of squamous genes (desmoglein 3 [*Dsg3*], involucrin [*Ivl*], and cornifin A [*Sprr1a*]) and a progressive decrease in the mRNA expression of *Krt5* across the CellState node assignments, which tracked with expression of the LSL-Apobec3 transgene (Fig. 2G, H and Supplementary Fig. 2H). Finally, to better define the CellState 4 population we performed differential gene expression between CellState 3 and CellState 4 in epithelial cells of UPPA tumors and found that many of the genes upregulated in CellState 4 were associated with squamous differentiation such as (*Cnfn, Sprr3, Sprr2d,* and *Sprr2i*) (Supplementary Fig. 2I). Moreover, GSEA (comparing CellState 3 and CellState 4) demonstrates a high level of enrichment of our squamous signature (Supplementary Fig. 2J). Overall, these findings are consistent with our hypothesis that CellState 4 cells are a highly squamous population.

## Apobec3 is sufficient to drive squamous differentiation and block luminal differentiation

The scRNA data provided correlative evidence that Apobec3 might be sufficient to induce squamous differentiation; however, we sought to understand this on a mechanistic level. To directly address whether mA3 can promote squamous differentiation, we engineered the murine urothelial carcinoma cell lines BBN963, BBN976, and UPFL.1 to express either doxycycline-inducible full-length mA3 (hereafter BBN963-mA3, BBN976-mA3, and UPFL.1-mA3) or doxycycline-inducible empty vector (hereafter BBN963-EV, BBN976-EV, and UPFL.1-EV)[21,24]. Additionally, we generated mA3-deaminase inactive BBN963 cells possessing an E73A mutation, which allows for maintenance of protein localization but abolishes the mA3 deaminase

function (Supplementary Fig. 3A, B) (hereafter BBN963-E73A). Following a 7-day induction of doxycycline, BBN963-mA3, BBN976-mA3, and UPFL.1-mA3 cells had significantly increased expression of *Dsg3* ($p < 0.01$), *Ivl* ($p < 0.001$), and *Sprr1b* ($p < 0.001$) as compared to empty vector expressing control cells and deaminase inactive BBN693-E73A cells (Fig. 3A and Supplementary Fig. 3C). In parallel, we generated organoids from *UPP* tumors (UPP-Org) with doxycycline-inducible full-length mA3 (UPP-Org-mA3). H&E staining showed the presence of keratin pearls in the UPP-Org-mA3 organoids but not the UPP-Org-EV (Fig. 3B).

To examine whether mA3-induced squamous differentiation can occur in normal mouse urothelial cells (or is only a property of transformed cells), we generated normal urothelial organoids with either doxycycline-inducible mA3 or empty vector (NUO-mA3 and NUO-EV). Similar to the findings in the mA3 cell lines, normal urothelial organoids overexpressing mA3 significantly upregulated *Dsg3* ($p < 0.001$), *Ivl* ($p < 0.001$), and *Sprr1b* ($p < 0.01$) as compared to EV controls (Fig. 3C). Immunofluorescence staining for DESMOGLEIN3 ($p < 0.0001$), INVOLUCRIN, and KERATIN6A in NUO-EV and NUO-mA3 showed significant increase in the expression of these squamous differentiation markers ($p < 0.0001$) (Fig. 3D, E).

To understand if APOBEC3 plays a role in not only squamous differentiation but also basal to luminal urothelial differentiation, organoids were cultured in media containing supplements to maintain proliferation or induce luminal differentiation. Work by the Clevers and Real groups, and recapitulated by our group, has demonstrated that organoid models maintain their proliferative capacity when cultured in media containing FGF7, FGF10, and A-83-01 (an ALK4/5/7 inhibitor) (proliferation media) and express high levels of basal markers such as *Krt5* and *Krt14*. In the absence of FGF7, FGF10, and A-83-01 (differentiation media), organoids undergo morphological and gene expression changes consistent with luminal differentiation[25,26] (Supplementary Fig. 3D, E). Using this system, we cultured NUO-EV and NUO-mA3 organoids in proliferation media for one week followed by differentiation media for an additional week. In differentiation media, NUO-EV organoids developed into lumen-containing organoids demonstrating phenotypic differentiation as well as upregulated expression of luminal markers *Upk1a* and *Upk3a* (Fig. 3F, G). However, NUO-mA3 organoids remained similar in appearance regardless of whether they were cultured in proliferative or differentiation media (Fig. 3F). While NUO-mA3 organoids upregulated *Upk1a* and *Upk3a* expression in differentiation media, the overall transcript levels were significantly lower than NUO-EV organoids grown in differentiation media ($p < 0.001$) (Fig. 3G). These data, in aggregate, suggest that expression of mA3 is sufficient to attenuate luminal differentiation in normal urothelial organoids and is consistent with the notion of mA3 maintaining a basal state while promoting a squamous phenotype.

## Activation of IL-1R1 signaling mediates mA3-induced squamous differentiation

We next sought to elucidate the signaling pathways downstream of mA3 activity that are responsible for the induction of the squamous transcriptional program. As APOBEC3 deaminase activity can result in cytosine mutations, we first looked to see if mA3-expressing cells had increased markers of DNA damage. To this end, we generated protein

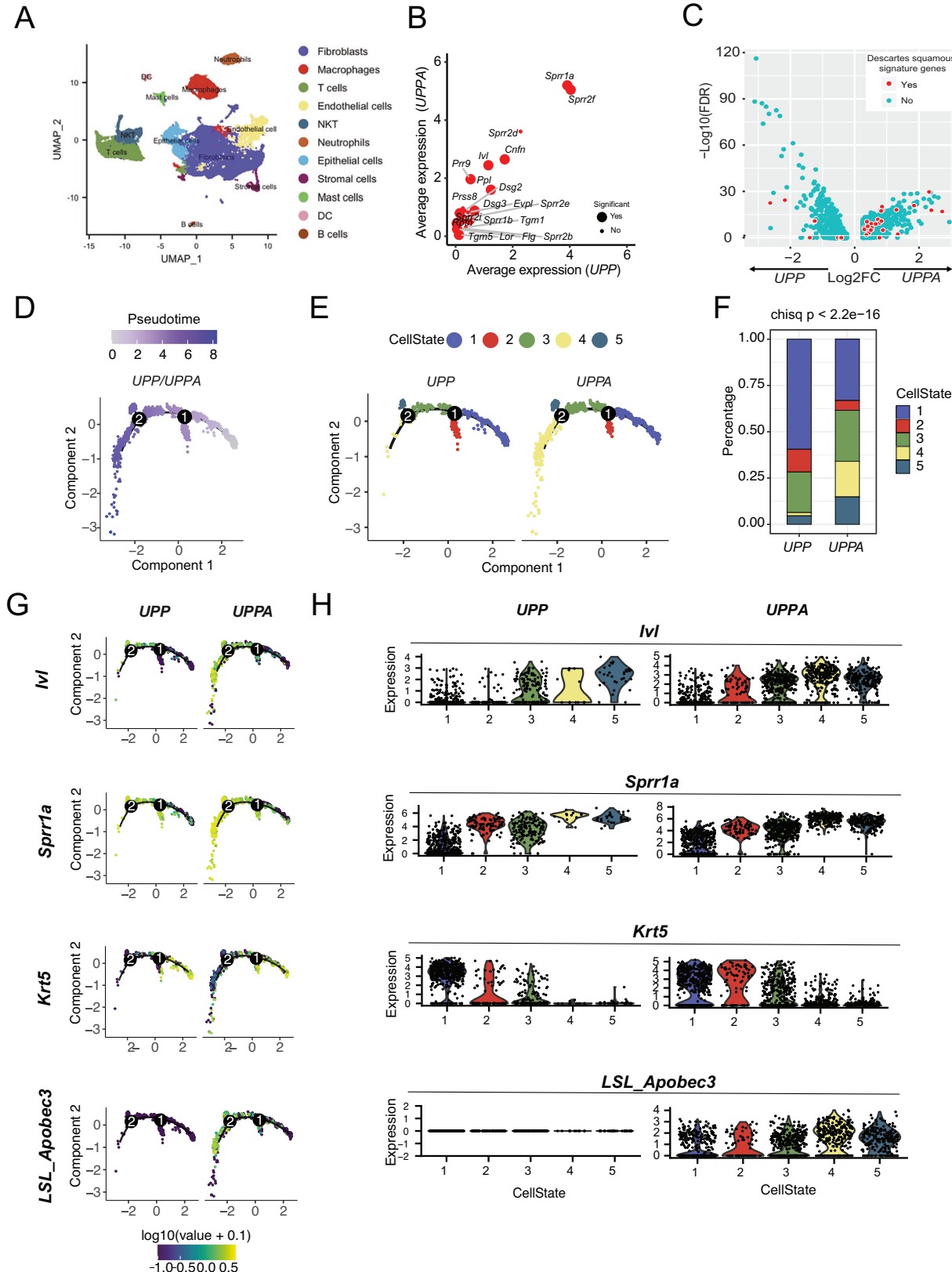

lysates from BBN963-mA3, BBN976-mA3, and UPFL.1-mA3 cells and performed western blot analysis for the DNA damage marker γH2A.X. Western blots showed a marked increase in γH2A.X protein in the mA3 expressing cell lines as compared to isogenic controls (BBN963-EV, BBN976-EV, and UPFL.1) after 1 week of doxycycline treatment (Fig. 4A and Supplementary Fig. 4A). As persistent DNA damage has been

shown to upregulate proinflammatory cytokine secretion[27], we then cultured parental BBN963 cells for 72 h in conditioned media from doxycycline treated BBN963-EV and BBN963-mA3. The mRNA expression of *Dsg3*, *Ivl*, and *Sprr1b* was significantly elevated in BBN963 cells ($p < 0.0001$) cultured in the mA3 conditioned media, but not the BBN963 cells cultured in EV conditioned media (Fig. 4B),

**Fig. 2 | scRNA-seq defines a population of tumor cells with a squamous gene expression profile enriched in UPPA tumors. A** UMAP plot of scRNA-seq data from UPP (n = 2) and UPPA (*n* = 2) tumors. **B** Dot plot showing the average expression of squamous markers in the entire epithelial cell population between UPP and UPPA tumors. Significance was calculated using the Wilcoxon rank-sum test followed by Bonferroni correction for multiple comparisons. **C** Volcano plot visualizing the Log2 Fold change of the differential gene expression analysis comparing the squamous clusters from UPP and UPPA tumors by Wilcoxon rank-sum. The color of the dot indicates if a gene belongs to the DESCARTES squamous epithelial cell signature. **D** Trajectory inference analysis of epithelial cell populations by plotted by pseudotime of UPP (n = 2) and UPPA (n = 2) tumors. **E** Trajectory inference analysis of epithelial cell populations plotted by cell state of UPP (n = 2) and UPPA (n = 2) tumors. **F** Stacked bar plot comparing the proportion of each CellState. Significance was calculated using a $\chi^2$ test, *p* < 2.2e$^{-16}$. **G** Trajectory inference analysis of epithelial cell populations indicated a linear progression from cells resembling basal cells (*Krt5*) to those more similar to differentiated squamous cells (*Sprr1a*) with high LSL_Apobec3 expression. **H** Violin plots showing the expression of the indicated genes between UPP and UPPA populations by cell state. See data availability statement for source data.

demonstrating that a secreted factor or factors from mA3 expressing cells could induce squamous gene expression.

We next wanted to understand what factors within the conditioned media may be mediating the upregulation of squamous genes. Prior studies have implicated a role for the inflammatory cytokine IL-1α in DNA damage response, as well as squamous trans-differentiation in pancreatic cancer[28]. We first compared the cellular expression of *Il1a and Il1b* mRNA between BBN963-EV and BBN963-mA3 cells. While both *Il1a* and *Il1b* were significantly upregulated by ectopic mA3 expression (*p* < 0.0001), *Il1a* had a 10x greater induction than *Il1b* (~50-fold vs ~5- fold) (Fig. 4C). The increase in transcription was manifested in an increased concentration of soluble IL-1α protein in the conditioned media from BBN963-mA3 cells, relative to BBN963-EV cells (*p* < 0.05) (Fig. 4D).

IL-1α is a cytokine expressed ubiquitously across different cell types in the body including epithelial cells, acting as an "alarmin" in response to cellular stress such as infection[29,30]. It has been shown that IL-1α plays a role in cellular response to DNA damage by UV light[31] and colocalizes to UV-induced DNA double-stranded breaks[32].

To test the sufficiency of IL-1α on inducing squamous gene expression, we treated parental BBN963 cells with recombinant IL-1α (rIL-1α) and found that IL-1α was sufficient to induce squamous gene expression (*Dsg3, Ivl* and *Sprr1b*) (*p* < 0.0001) (Fig. 4E). Moreover, we examined whether recombinant IL-1α was sufficient to induce histologic squamous differentiation. To this end, we treated *UPP* tumor organoids with rIL-1α for 1 week, followed by H&E staining. IL-1α-exposed *UPP* tumor organoids showed the presence of squamous differentiation characterized by an increased amount of cytoplasm and evidence of early keratin pearl formation (Fig. 4F, arrow). rIL-1β was also sufficient to induce squamous gene expression (Supplementary Fig. 4B). Next, we asked whether the interleukin-1 receptor (IL-1R) signaling is necessary for the squamous gene induction. To this end, we utilized both an anti-IL-1α neutralizing antibody and CRISPR knock-out of *Il1r1* in parental BBN963 cells (Supplementary Fig. 4C). In both approaches, blocking Il-1r signaling abrogated the induction of squamous markers in BBN963 cells treated with mA3 conditioned media (*p* < 0.0001) (Fig. 4G, H). Together with our prior results, this indicates that IL-1α and IL-1R signaling are both necessary and sufficient for mA3-induced squamous differentiation.

Interestingly, while having no deaminase activity (Supplementary Fig. 3B), BBN963-E73A showed a comparable level of increase in γH2A.X compared to BBN963-mA3 cells (Fig. 4A). Despite this increase in γH2A.X, BBN963-E73A cells did not promote squamous gene expression (Fig. 3A) and only an intermediate amount of IL1A (Fig. 4D). The ability of APOBEC3 to induce DNA damage in a deaminase independent manner has been shown by previous groups[33–36]. Taken together, these results suggest that the DNA damage induced by mA3 is independent of its deaminase function, but also suggest that induction of squamous differentiation is more closely linked to the induction of IL1A.

### cFOS is necessary for mA3-induced squamous differentiation and suppresses luminal transcription factors
Canonical downstream signaling from IL-1R is mediated either through the activation of p38, ERK and JNK kinases or through activation of

NF-κB, ultimately leading to dimerization of FOS and JUN to form the AP-1 transcription factor. We examined if one or both pathways was upregulated by mA3 expression and found that the levels of p-p38, p-ERK1/2, and p-cFOS were all higher in BBN963-mA3 relative to BBN963-mE73A and BBN963-EV (Fig. 5A, B and Supplementary Fig. 5A). In contrast, we found no change in the canonical NF-κB pathway signaling proteins in BBN963-mA3 cells as evidenced by invariant levels of p-IKKα/β and TNFAIP3, a transcriptional target of NF-κB, and a small decrease in the level of p-p65, (Fig. 5C).

Because cFOS has been implicated in promoting squamous differentiation[37–39], we next asked whether activation of cFOS by mA3 was dependent upon IL-1α. Indeed, we found that treating BBN963-EV and BBN963-mA3 with an anti-IL-1α neutralizing antibody could blunt mA3 induction of p-cFOS (Fig. 5D). Moreover, to determine whether cFOS is necessary for mA3 dependent induction of squamous gene expression, we infected CRISPR Cas9 expressing BBN963-EV and BBN963-mA3 cells with either sgSCR_control (scrambled) or sgFos (FBJ murine osteosarcoma viral oncogene homolog) (Supplementary Fig. 5B). While the sgSCR_control did not alter squamous gene expression, infection with sgFOS blunted squamous gene expression in the BBN963-mA3 cells, consistent with the notion that mA3 promotes squamous differentiation in part through cFOS (Fig. 5E). In aggregate these data demonstrate that mA3-induced DNA damage triggers IL-1α production that engages IL-1R in an autocrine or paracrine manner to promote p38-ERK signaling and downstream AP-1 transcription factor activation and drives squamous gene expression and differentiation.

### Apobec3 overexpression suppresses FOXA1, GATA3, and PPARγ expression
Considering our data suggesting IL-1α is signaling through cFOS, we asked if our earlier data demonstrating that mA3 blocks luminal differentiation could also be in part explained by this signaling axis. The key transcription factors that drive differentiation of urothelial basal cells to superficial umbrella cells have been well delineated and include: FOXA1, GATA3, and PPARγ. These same factors have also been shown by our lab and others to be defining features of the luminal molecular subtype of bladder cancer[6,40,41]. The Mendelsohn group has demonstrated the importance of PPARγ in restricting squamous differentiation[42]. Moreover, Warrick et al. demonstrated that the squamous regions in UC tumors with squamous differentiation have a marked reduction in the expression of FOXA1, GATA3 and PPARγ compared to urothelial regions[5]. We therefore asked whether *Apobec3* overexpression could modulate the expression of these luminal- defining transcription factors and, if so, does sgFOS reverse this. We found that FOXA1, GATA3 and PPARγ protein levels were markedly reduced in BBN963-mA3 cells (Fig. 5F–H), and this effect was to some extent independent of deaminase activity, as PPARG and GATA3, but not FOXA1, were downregulated in mA3-E73A expressing BBN963 cells (Supplementary Fig. 5C). Gene signatures of PPARG and GATA3 activation[43] were also downregulated when assessed in paired areas of UC and squamous differentiation (Fig. 5I). We next explored if mA3's effect on GATA3 and PPARγ is cFOS dependent. We found that *FOS* CRISPR increased the protein levels of GATA3 and PPARγ in BBN963-mA3 cells (Fig. 5J).

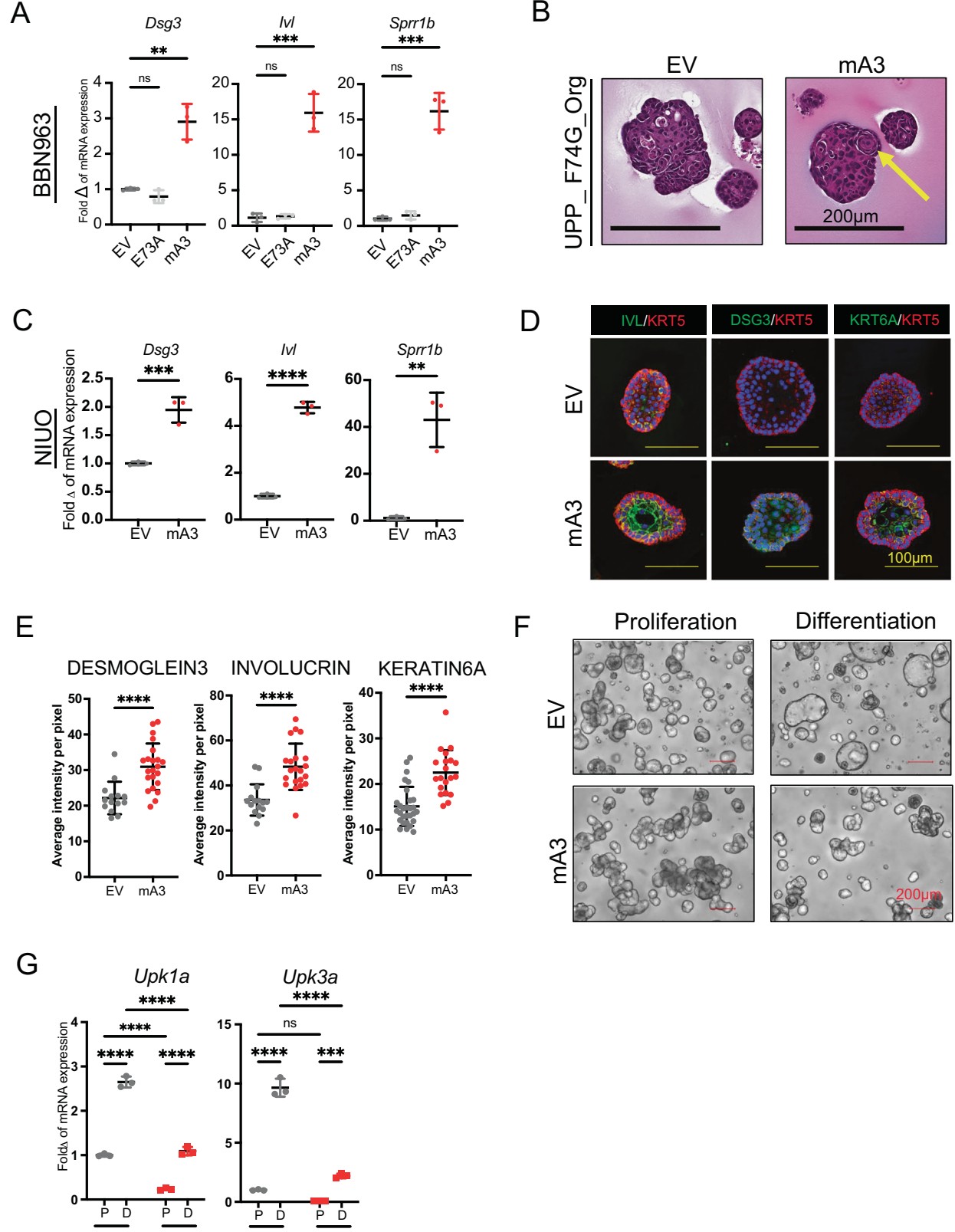

## Human *APOBEC3A* but not *APOBEC3B* is associated with squamous differentiation

While our work demonstrates the ability of murine Apobec3 to promote squamous differentiation in vitro and in vivo, the transfer of these findings to humans is more complex. The human genome contains seven independent *APOBEC3* genes. In an effort to understand

which, if any, APOBEC3 enzymes may have a similar influence on squamous differentiation, we used gene expression data from The Cancer Genome Atlas (TCGA BLCA) to correlate expression of each APOBEC family member with the expression of a panel of squamous genes. Of the 7 family members, only *APOBEC3A* (*A3A*) was consistently correlated to squamous gene expression (Fig. 6A). Similar

**Fig. 3 | Apobec3 drives squamous differentiation. A** Scatter plot showing the expression of squamous markers (*Dsg3*, *Ivl*, and *Sprr1b*) in BBN963-EV, BBN963-E73A, and BBN963-mA3 cells after 7 days of doxycycline treatment. Significance was calculated using two-way ANOVA followed by Tukey's multiple-comparison test. Data represent mean ± SD ($n = 3$, technical replicates), **$p < 0.01$, ***$p < 0.001$, ns not significant. **B** Representative H&E images showing the presence of keratin pearls in mA3 expressing UPP_F74G_Org. **C** Scatter plots showing the expression of squamous markers in organoids NUO-EV and NUO-mA3 after 7 days of doxycycline treatment. Significance was calculated using a two-sided unpaired *t*-test. Data represent mean ± SD ($n = 3$, technical replicates), **$p < 0.01$, ***$p < 0.001$, ****$p < 0.0001$. **D** Immunofluorescence images of NUO-EV and NUO-mA3 organoids stained for the indicated squamous markers after 7 days of doxycycline treatment. **E** Quantification of fluorescence intensity of organoids stained for the squamous differentiation markers in (**D**). Each point represents a distinct organoid. Data represent mean ± SD. Significance was calculated using a two-sided Mann−Whitney *U*-test, ****$p < 0.0001$. **F** Brightfield images of NUO-EV and NUO-mA3 organoids under proliferation and differentiation conditions. **G** Scatter plots showing the expression of luminal genes in NUO-EV and NUO-mA3 organoids under proliferation (P) and differentiation (D) conditions. Significance was calculated using two-way ANOVA followed by Tukey's multiple-comparison test. Data represent mean ± SD ($n = 3$, technical replicates), ***$p < 0.001$, ****$p < 0.0001$, ns not significant. Source data are provided as a Source Data file.

findings were validated in the UC-GENOME cohort (Supplementary Fig. 6A).

To better understand if a specific cell type was responsible for *APOBEC3A* expression, we examined scRNAseq data from five previously unpublished UC tumors from our own institution ($n = 4$ pure UC, $n = 1$ UC with 5% squamous differentiation) as well as scRNAseq from a single tumor from Warrick et al. derived from a urothelial tumor with extensive squamous differentiation[5]. We calculated the correlation between the expression of APOBEC3 family members and our squamous gene signature within epithelial cells. Again, *APOBEC3A* had the highest correlation to regions/tumors with squamous differentiation (Fig. 6B).

Next, scRNA-seq data from Warrick et al., was clustered and cell type prediction was performed using SingleR. On the UMAP 1 axis, a subset of cells, consisting of 2 clusters, was shifted to the right and away from the other clusters. SingleR-based cell type prediction identified these two clusters as epithelial cells (cluster 12) and keratinocytes (cluster 2), consistent with gene expression patterns seen in the scRNA-seq data (Supplementary Fig. 6B). Similar to our findings in mice, *IL1A* is expressed at increased levels within cluster 2 cells that also expressed *A3A* and squamous marker genes (Fig. 6C) but not other APOBEC3 family members (Supplementary Fig. 6C). Warrick et al. had also performed bulk RNAseq from paired regions of UC with squamous differentiation. Examination of these data also demonstrated that squamous regions had significantly increased *APOBEC3A* expression ($p < 0.019$) as compared to the paired UC regions; however, *APOBEC3B* (A3B) ($p < 0.13$) and other APOBEC3 family members were invariant (Fig. 6D and Supplementary Fig. 6D).

To see if the association of squamous differentiation with A3A but not A3B was consistent across larger cohorts, we analyzed tumors from The Cancer Genome Atlas Bladder Cancer (TCGA BLCA), UC-GENOME, and the TCGA PanSquamous studies[7,44,45]. In TCGA BLCA and UC-GENOME, tumors that were noted by pathologists as having histologic squamous differentiation had significantly increased *A3A* but not *A3B* gene expression ($p = 3.2 \times 10^{-4}$, $p = 0.19$, respectively) as compared to tumors without noted squamous differentiation (Fig. 6E and Supplementary Fig. 6E). We next extended this analysis to Consensus molecular subtypes and saw that there were significant differences between *A3A* and *A3B* expression, with Basal/Squamous tumors having increased levels of *A3A*, while LumU was highest for *A3B* ($p < 1 \times 10^{-5}$) (Fig. 6F and Supplementary Fig. 6F)[46]. Additionally, this pattern was not restricted to only bladder cancer. Data from TCGA PanSquamous study, which used clustering to define molecular features of squamous tumors, showed increased expression of *A3A* but not *A3B* in tumors that were identified molecularly as squamous, regardless of tumor type (Supplementary Fig. 6G).To validate that hA3A was sufficient to induce DNA damage, as assessed by increased γH2A.X, we ectopically expressed hA3A in BBN963 and UPFL.1 cells, which demonstrated deaminase activity (Supplementary Fig. 7A). Concomitant with A3A overexpression, we observed an increase in γH2A.X protein (Fig. 6G and Supplementary Fig. 7B) and increased expression of squamous gene markers (Fig. 6H and Supplementary Fig. 7C).

While scRNA-seq and multi-regional sequencing has demonstrated the co-expression of *APOBEC3A* and squamous marker genes, we wanted to directly visualize the co-expression within morphologically squamous cells. To accomplish this, we performed Xenium spatial transcriptomic analysis using a custom 477 gene panel (Supplementary Table 2) on an FFPE section from the same UC tumor that contained 5–10% squamous differentiation as in Fig. 6B. Using the Xenium Explorer software package, the tissue sample was pseudo-colored by the kmeans = 10 clustering solution and aligned with a hematoxylin and eosin stained serial section of the tumor (Fig. 7A). A region of interest (ROI) containing squamous differentiation was identified and annotated by a fellowship trained GU pathologist (Fig. 7B). Transcript density for *APOBEC3A*, *IVL* and *KRT6A* was then visualized and transcript numbers quantified for the two predominate cell clusters within the ROI. Both *APOBEC3A* and *IVL* had significantly increased expression in cells belonging to cluster 4 ($p < 0.01$), while *KRT6A* was universally high within all epithelial cells, but enriched within Cluster 2 (Fig. 7C).

To better understand the distribution of cell types across the sample and identify if other epithelial clusters were present, the per-cluster mean expression for a subset of cell-type-specific marker genes was calculated, and hierarchical clustering was performed. Based on marker gene expression, Cluster 2 (*KRT5*, *DSG3*, *KRT6*), Cluster 4 (*KRT14*, *APOBEC3A*, *IVL*, *DSG3*, *KRT6A*) and Cluster 5 (*KRT20*, *UPK3B*) were identified as epithelial cluster, corresponding to basal-like, squamous, and luminal cells, respectively (Fig. 7D). In a subset analysis of the epithelial cells, Xenium data was consistent with our bulk and single cell analysis, showing that *APOBEC3A* and not *APOBEC3B* was enriched in the squamous population (Fig. 7C). Squamous cells that expressed both *IVL* and *KRT6A* were more likely to co-express *APOBEC3A* than *APOBEC3B* (Fig. 7E). Altogether, the human data and murine overexpression model, strongly implicate *APOBEC3A* as the APOBEC3 family member driving squamous differentiation in UC[47].

## Discussion

Analysis of somatic mutations in MIBC demonstrates a striking contribution of APOBEC-induced mutagenesis[48]. Subsequent reports have established that APOBEC mutational patterns are also present in NMIBC[49,50]. While elevated APOBEC mutational load has been associated with a better prognosis and response to immunotherapy in bladder cancer[7,44,51], precisely how APOBEC3 overexpression and mutagenesis functionally impact bladder cancer biology has been underexplored. While the roles of *APOBEC3A* and *APOBEC3B* in bladder cancer have not been functionally explored, transgenic overexpression of *APOBEC3G* in a BBN-induced carcinogen mouse model of bladder cancer has documented that *APOBEC3G* can shorten survival as well as drive genomic instability[19]. Additionally, the role of the APOBEC3 family in tumorigenesis has been explored by Harris and colleagues in the context of murine models of colon and liver cancer. Law et al. demonstrated that transgenic overexpression of A3A but not A3G promoted colon polyp formation in the Apc^Min model of colon carcinogenesis[13]. Moreover, in the fumarylacetoacetate hydrolase (FAH) liver carcinogenesis model, where all seven APOBEC3 family

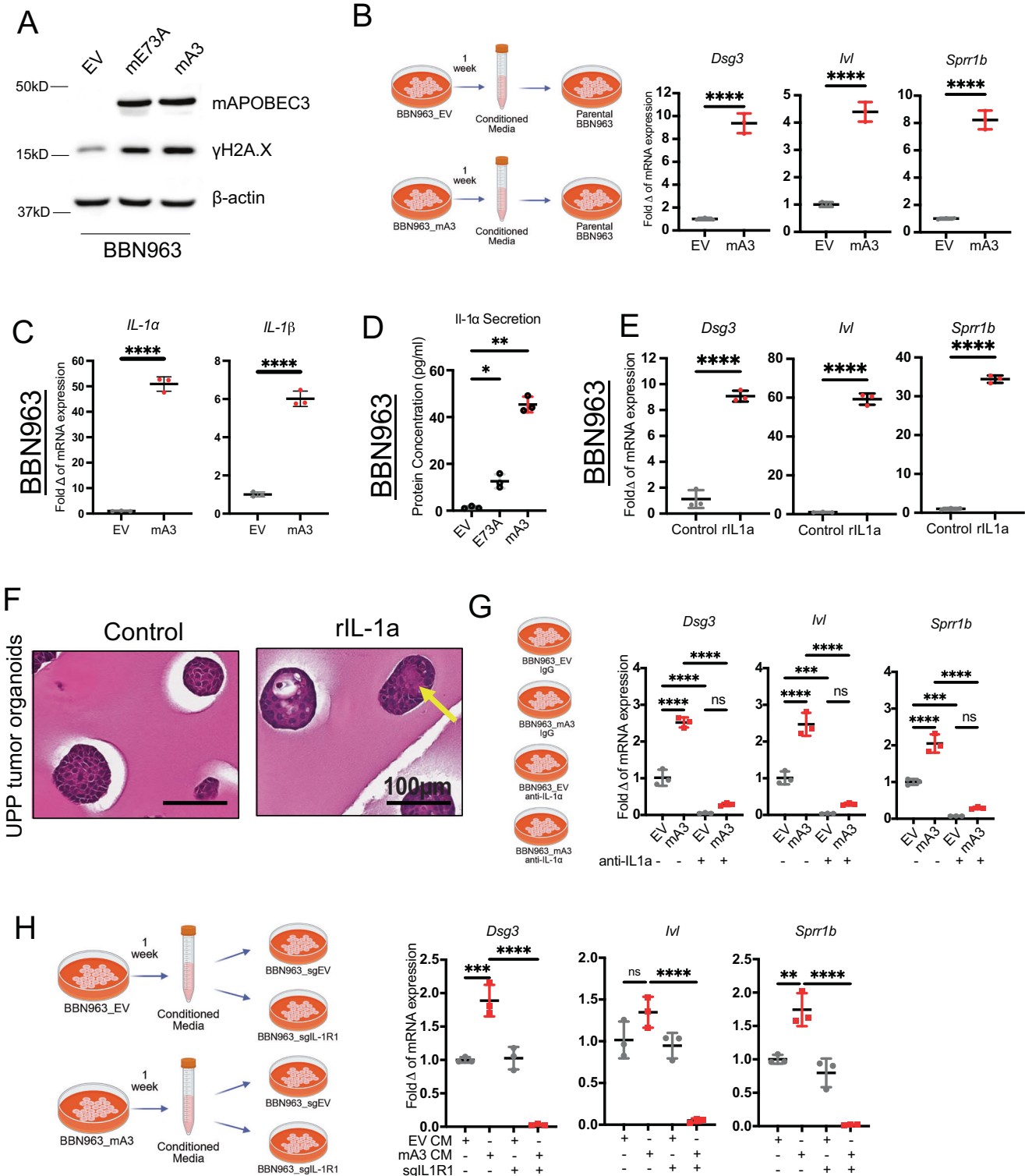

members were tested, only A3A promoted an increased frequency of liver tumors in mice[13]. In keeping with these studies, our work demonstrates that mA3 plays a functional role in driving bladder tumor progression in mice and promoting squamous trans-differentiation[52].

Our *Rosa26^LSL-Apobec3* knock-in mouse leverages the fact that mice have a single Apobec3 family member and allows us to query APOBEC3 biology with a single allele. Additionally, mA3 and human APOBEC3A and APOBEC3B only have 36 and 33% amino acid identity, respectively, and thus our use of mA3 avoids any issues of exogenous expression of an allo-antigen for future immunologic studies.

We found that mA3 had profound phenotypic consequences, such as promoting squamous differentiation and in the pseudotime analysis from our scRNAseq of UPP and UPPA tumors, noted cell states of highly squamous cells (cell states 4 and 5) in UPPA tumors. We mapped reads to the LSL-Apobec3 cassette, which allowed us to lineage trace any cells that had undergone Cre-recombination. Interestingly, we found that there were LSL-Apobec3-expressing cells throughout the continuum of cell states, suggesting that Apobec3 expression does not immediately promote squamous differentiation. Indeed, this late-onset squamous induction could result from several

**Fig. 4 | Activation of IL1A/AP-1 signaling pathway mediates Apobec3-induced squamous differentiation. A** Western blots of whole cell lysates from BBN963-EV, BBN963-E73A, and BBN963-mA3 treated with doxycycline for 7 days blotted with indicated antibodies. **B** Scatter plots showing the expression of squamous markers in parental BBN963 cells treated for 24 h with conditioned media collected from BBN963-EV and BBN963-mA3 cells after 7 days of doxycycline treatment. Significance was calculated using a two-sided unpaired $t$-test. Data represent mean ± SD ($n = 3$, technical replicates), ****$p < 0.0001$. Schematic Created in BioRender. Sturdivant, M. (2025) https://BioRender.com/ddrl084. **C** Scatter plots showing the expression of proinflammatory cytokines (IL-1α and IL-1β) in BBN963 treated with doxycycline for 7 days. Significance was calculated using a two-sided unpaired $t$-test. Data represent mean ± SD ($n = 3$, technical replicates), ****$p < 0.0001$. **D** Scatter plot showing the expression of secreted IL-1α in BBN963-mA3, BBN963-E73A, and BBN963-EV cells via ELISA assay. Significance was calculated using two-way ANOVA followed by Tukey's multiple-comparison test. Data represent mean ± SD ($n = 3$, technical replicates), *$p < 0.05$. **E** Scatter plots showing the expression of squamous markers in parental BBN963 cells treated with recombinant IL-1α for 72 h.

Significance was calculated using a two-sided unpaired $t$-test. Data represent mean ± SD ($n = 3$, technical replicates), ****$p < 0.0001$. **F** Representative H&E images of UPP tumor organoids treated with rIL-1α for 1 week. **G** Scatter plots showing the expression of squamous markers in BBN963-EV and BBN963-mA3 treated with doxycycline with or without anti-IL1a neutralizing antibody for 7 days. Significance was calculated using two-way ANOVA followed by Tukey's multiple-comparison test. Data represent mean ± SD ($n = 3$, technical replicates), ***$p < 0.001$, ****$p < 0.0001$, ns not significant. Schematic created in BioRender. Sturdivant, M. (2025) https://BioRender.com/9migzjm. **H** Scatter plots showing the expression of squamous markers in BBN963-sgEV and BBN963-sgIL1R1 cells treated for 24 h with conditioned media collected from BBN963-EV and BBN963-mA3 cells after 7 days of doxycycline treatment. Significance was calculated using two-way ANOVA followed by Tukey's multiple-comparison test. Data represent mean ± SD ($n = 3$, technical replicates), **$p < 0.01$, ***$p < 0.001$, ****$p < 0.0001$, ns not significant. Schematic created in BioRender. Sturdivant, M. (2025) https://BioRender.com/08mppjc. Source data are provided as a Source Data file.

mechanisms, including the chronic accumulation of Apobec3-induced DNA damage to induce sufficient cytokine production prior to squamous induction or the kinetics of basal/luminal transcription factor downregulation.

Of the seven human APOBEC3 family members, our in silico data strongly associate A3A with squamous differentiation. Indeed, we saw that A3A was the only APOBEC family member whose expression correlates with squamous differentiation genes in bulk RNA-seq, but whether A3A expression is coming from tumor cells or other cells within the tumor microenvironment is unclear. We leveraged the power of scRNA-seq and spatial transcriptomics to specifically discern that A3A is expressed in tumor cells and is correlated with expression of our squamous gene signature on a single-cell level. Finally, we found that overexpression of A3A in BBN963 cells was sufficient to induce squamous gene expression. Therefore, A3A is uniquely associated with squamous differentiation.

Why only A3A is associated with squamous differentiation is unclear, as all the human APOBEC3 enzymes can induce DNA damage to some degree. It has been established in other studies that APOBEC3A likely causes more DNA damage than the other APOBEC3 enzymes, which potentially plays an important role in their ability to promote squamous differentiation[53]. However, it remains interesting to hypothesize whether there are additional mechanisms underlying this phenotype. For instance, similar to mouse Apobec3, several human APOBEC3 enzymes reside in both the cytoplasm and the nucleus, including APOBEC3A, whereas human APOBEC3B resides predominantly in the nucleus[54]. There has been evidence that mitochondrial DNA (located in the cytoplasm) can serve as the substrate for APOBEC3A[55,56]. Therefore, of all human APOBEC3 enzymes, APOBEC3A stands out as the only one that can concomitantly potently catalyze cytosine deamination in both nuclear and mitochondrial DNA. We postulate that this distinctive feature of APOBEC3A is what makes it the only APOBEC3 enzyme that can drive squamous differentiation. As many bladder cancers have evidence of APOBEC-induced mutagenesis, it is interesting that only 30% of urothelial carcinomas have squamous differentiation. There may be multiple mechanisms to account for this, including the possibility that APOBEC-mutagenesis (and squamous differentiation) is episodic, as seen in cancer cell lines[47], as well as that there are multiple mechanisms for UC squamous differentiation. We have shown that IL-1A is sufficient to induce squamous differentiation, therefore, alternative APOBEC-independent pathways that upregulate IL1A may also be sufficient to promote squamous differentiation.

Squamous differentiation has been recognized as a manifestation of lineage plasticity and as a mechanism of therapy resistance to both EGFR and ALK inhibitors in lung adenocarcinomas[57–60] as well as KRAS G12C inhibitors (adagrasib)[61]. Thus, inhibiting lineage plasticity and

urothelial to squamous trans-differentiation could potentially improve the efficacy of treatment or stave off therapeutic resistance. While the development of APOBEC3 inhibitors is underway, they remain at an early stage. Notably, our work defines IL-1α - AP-1 signaling as a key downstream mediator of APOBEC3-induced squamous differentiation. Disrupting IL-1α, which is highly targetable, might therefore be sufficient to block squamous trans-differentiation. Several agents are either approved or under clinical development to target the IL-1α signaling pathway. For example, anakinra is an FDA-approved recombinant IL-1R1 antagonist used to treat rheumatoid arthritis, neonatal onset multi-system inflammatory disease, and deficiency of interleukin-1 antagonist. MEDI8968 and AMG 108 are two fully humanized monoclonal antibodies targeting IL-1R1 that have been investigated for chronic obstructive pulmonary disease and osteoarthritis, respectively. Bermekimab is a human antibody targeting IL-1α that was studied in a clinical trial for advanced tumors. One concern is whether IL-1 blockade could promote phenotypic shift from a basal/squamous tumor to a more luminal UC phenotype, thus promoting a tumor that is less responsive to chemotherapy. We previously pointed to the molecular parallels between breast and bladder cancer and suggested that molecular subtype might guide therapy stratification[40]. While in breast cancer it is well established that basal-like tumors (in contrast to luminal) have meaningful clinical benefit from chemotherapy, the findings in bladder cancer are less clear. Early studies[62] suggested that, akin to breast cancer, only basal bladder tumors had a clinical benefit to NAC. Subsequent studies have challenged this notion. Indeed, recent data from the Lund group suggest that luminal tumors of the Uro (urothelial-like) and GU (genomically unstable) subtypes derive a similar response to cisplatin as the Ba/Sq subtype[63]. Therefore, while targeting IL-1α is therapeutically tractable, the ultimate impact remains to be seen.

In summary, our work defines a key role of APOBEC3 enzymes in driving squamous trans-differentiation of bladder cancer. We demonstrate the necessity of IL-1α/AP-1 signaling for these APOBEC3-dependent phenotypes. These studies in faithful bladder cancer GEM models demonstrate that APOBEC3 promotes cancer cell progression, and the scRNA-seq from human tumors implicates human APOBEC3A as the causative driver of squamous differentiation. The upregulation of APOBEC3 enzymes has been demonstrated in other malignancies such as breast, lung, cervical and head & neck cancers. Our findings can potentially be applied to these cancer types to understand the mechanisms that promote aggressive disease in these settings.

## Methods
### Mouse Apobec3 model
A Cas9 guide RNA targeting the mouse *Rosa26* first intron (protospacer sequence 5'-GGAGTTGCAGATCACGA-3') was cloned into a *T7* promoter vector, followed by *T7* in vitro transcription and spin column

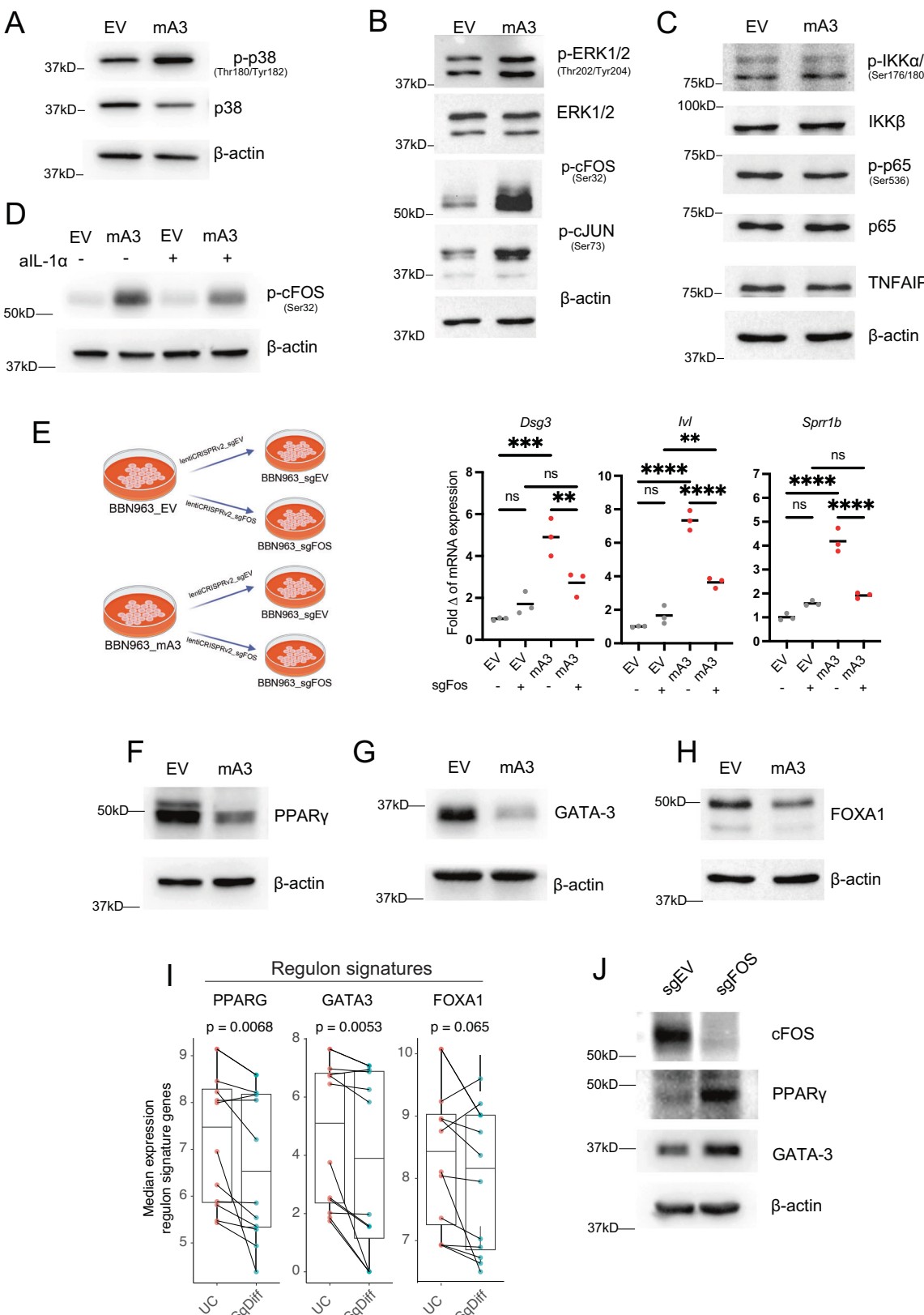

purification, with elution in microinjection buffer (5 mM Tris-HCl, pH 7.5, 0.1 mM EDTA). The donor plasmid included *Rosa26* gene homology arms flanking a neomycin resistance cassette, *CAG* promoter, *LSL* element, *mApobec3* cDNA (BC003314), Woodchuck Hepatitis Virus Posttranscriptional Regulatory Element (WPRE), and rabbit beta-globin polyadenylation sequence. The donor vector was prepared by the Qiagen High Speed Maxiprep protocol and resuspended in microinjection buffer. Recombinant Cas9 protein was expressed in E. coli and purified by the UNC Chapel Hill Protein Expression and Purification Core Facility. C57BL/6J two-cell embryos were micro-injected with 400 nM Cas9 protein, 25 ng/µl guide RNA and 20 ng/µl donor plasmid in microinjection buffer. Injected embryos were

**Fig. 5 | Activation of the AP-1 transcription factor mediates Apobec3-induced squamous differentiation. A–C** Western blots of whole cell lysate from BBN963-EV and BBN963-mA3 cells treated with doxycycline for 7 days, blotted for the indicated antibodies. **D** Western blots of whole cell lysates from BBN963-EV and BBN963-mA3 cells treated with doxycycline, with or without anti-Il1a neutralizing antibodies for 7 days. **E** Scatter plots showing the expression of squamous markers in BBN963-EV and BBN963-mA3 cells with Fos CRISPR after 7 days of doxycycline treatment. Significance was calculated using two-way ANOVA followed by Tukey's multiple-comparison test. Data represent mean ± SD ($n = 3$, technical replicates), **$p < 0.01$, ***$p < 0.001$, ****$p < 0.0001$, ns not significant. Schematic created in BioRender. Sturdivant, M. (2025) https://BioRender.com/34m5tr5. **F–H** Western blots of whole cell lysate from BBN963-EV and BBN963-mA3 cells treated with doxycycline for 7 days, blotted for indicated antibodies. **I** Box plots of median expression of indicated gene signatures[43] from bulk RNA-seq of paired UC and squamous differentiation regions from Warrick et al. The center line plotted in the box represents the median, the upper and lower bounds of the box represent the 75th and 25th percentiles, respectively, and the whiskers indicate the minima and maxima values. The significance was calculated using a two-sided paired Wilcoxon signed rank-sum test. **J** Western blots of whole cell lysate from BBN963-sgEV and BBN963-sgFOS cells treated with doxycycline for 7 days, blotted for indicated antibodies. Source data are provided as a Source Data file.

implanted in recipient pseudopregnant females. Resulting pups were screened by PCR for the presence of the correct knock-in event. The founders were mated to wild-type C57BL/6J females, and the resulting pups were screened with the same assays used to detect correct integration events in the founders.

The *Rosa26-LSL-mApobec3* mice were then crossed with our UPPL model[21], (B6.129P2-*Trp53*tm1Brn/J [Jackson Lab, stock: 008462], *Pten* conditional knockout mice [obtained from Terry Van Dyke], B6;DBA-Tg[*Upk3a*-GFP/cre/ERT2]26Amc/J [Jackson Lab, stock: 015855], and FVB.129S6[B6]-Gt[ROSA]26Sortm1[Luc]Kael/J [Jackson Lab, stock: 005125] then backcrossed to C57BL/6 (in house breeding). The resulting UPPA and UPP genotypes were then maintained and used in all in vivo studies.

### Animal studies
For all in vivo studies involving the use of either the UPP or UPPA model, 8–10-week-old mice were treated with tamoxifen (5 mg) by oral gavage every other day for a total of three (3) doses to induce Cre recombinase activity. Mice were regularly monitored for tumor growth by bladder ultrasonography. Mice were sacrificed for the humane endpoints when weight loss was more than 10% of the initial weight or tumor size reached a diameter of >7 mm as evaluated by bladder ultrasound, as permitted by the IACUC protocol. All mice were housed under a 12-h light cycle and at an ambient temperature of 68–74 °F (30–70% humidity).

### Doxycycline-inducible Apobec3
*mApobec3* cDNA plasmid was purchased from Horizon/PerkinElmer (MMM1013-202761182). The catalytic-dead *mApobec3* was generated through site-directed mutagenesis, performed by Azenta Life Sciences, to create the E73A mutation. *hApobec3a* cDNA plasmid was purchased from GenScript (OHu18813). The cDNA was cloned into pInducer20-Blast, a gift from Dr. Jean Cook (Addgene, 109334).

### Adherent cell culture
Mouse bladder cancer cell lines BBN963, BBN976, and UPFL.1[21,24] were transduced to express doxycycline-inducible Apobec3. The UPFL.1 cell line had the endogenous mAPOBEC3 removed through CRISPR knockout. All cell lines were cultured in DMEM (Gibco) supplemented with 10% fetal bovine serum, and 1X penicillin/streptomycin (Gibco). All cell lines were maintained at 37 °C in 5% CO$_2$. Mycoplasma testing was performed routinely while cells were in culture. To induce the expression of mApobec3, doxycycline was added to the culture media at 4 μg/ml. For a 7-day induction, fresh media with doxycycline were replaced every 2–3 days. To block IL1a, anti-IL1a neutralizing antibody (Biolegend, clone ALF-161) was used at 2 μg/ml. Cells were treated with cytokines at 100 ng/mL. The conditioned media were collected by centrifuging and passing the media through a 0.45-μm filter to remove floating cells and debris.

### Establishment and maintenance of organoid culture
Bladder tissue was excised and minced gently with a scalpel under sterile conditions. The tissue was then incubated in 1 ml solution of 1 mg/ml collagenase (Sigma-Aldrich, C9891) dissolved in Advanced DMEM/F12 (Gibco, 1263410) supplemented with 1X Antibiotic-Antimycotic (Gibco, 15240062) and 10 μM Y-27632 (ApexBio, A3008) for an hour at 37 °C on a rocker. The enzymatically digested tissue was then passed through a 70-μm filter and centrifuged at $193 \times g$ for 5 min. The cell pellet was resuspended in 1 mL of DMEM (Gibco, 10313020) supplemented with 10% FBS and 1X Penicillin-Streptomycin (Gibco, 15140122) and transferred into a 1.5 mL Eppendorf tube. The cell suspension was centrifuged, and the cell pellet was finally resuspended in 100 μL of Matrigel (Corning, 356231). Cell suspension in Matrigel was pipetted into a 24-well plate to form a dome in the center of the well. The plate was transferred into the 37 °C incubator to allow the Matrigel to polymerize for 30 min. 500 μL of organoid growth media consisting of Advanced DMEM/F12, 1X Antibiotic-Antimycotic, 1X B-27 supplement (Gibco, 17504044), 100 ng/ml FGF10 (PeproTech, 100-26), 25 ng/mL FGF7 (PeproTech, 100-19), 500 nM A-83-01 (Sigma-Aldrich, SML0788), 10 μM Y-27632 was added to each well. Organoid growth media was refreshed every 2–3 days.

To differentiate the organoids, the organoids were cultured in growth media for -1 week. Then differentiation media consisting of Advanced DMEM/F12, 1X Antibiotic-Antimycotic, 1X B-27 supplement was then used to culture and differentiate the organoids for an additional 1 week.

### CRISPR/cas9 knockout
The RNA guides were designed using Broad Institute sgRNA Designer. The guides were cloned into the lentiCRISPRv2 puro vector. The lentiCRISPRv2 puro was a gift from Brett Stringer (Addgene, plasmid 98290).

### Western blot
Cells were lysed in 1× RIPA buffer (Abcam) supplemented with protease inhibitor cocktail (Roche) and phosphatase inhibitor cocktail (Calbiochem). Cell extracts (20–30 μg) were resolved by SDS-PAGE, transferred to a nitrocellulose membrane, and then probed with the indicated antibodies (Supplemental Table 3). Proteins were visualized with the SuperSignal™ West Femto Maximum Sensitivity Substrate (Thermo Scientific) using HRP-conjugated anti-rabbit (catalog 31462) or anti-mouse (catalog 31432) secondary antibodies (Thermo Scientific).

### Deaminase assay
Cells were lysed in EBC lysis buffer (50 mM Tris-HCl, pH 7.5, 120 mM NaCl, and 0.5% NP-40) supplemented with protease inhibitor cocktail (Roche) and phosphatase inhibitor cocktail (Calbiochem). About 10 μl of cell extract (10μg) were incubated with 10 μl of deaminase reaction mix[64] with the oligonucleotides (Integrated DNA Technologies) listed below for 2 h at 37 °C. Then 1 N NaOH (μl) was added to the reaction mix and incubated at 95 °C for 10 min. About 30 μl of 2x RNA loading dye (New England Biolabs) was then added to the mixture and incubated at 95 °C for 2 min. About 5 μl of reaction mixture was loaded into a 15% TBE-Urea Gel (pre-ran at 150 V for 1 h) and run at 150 V for 1 h. Gels were imaged with a Typhoon FLA 9500 imager and analyzed in ImageJ

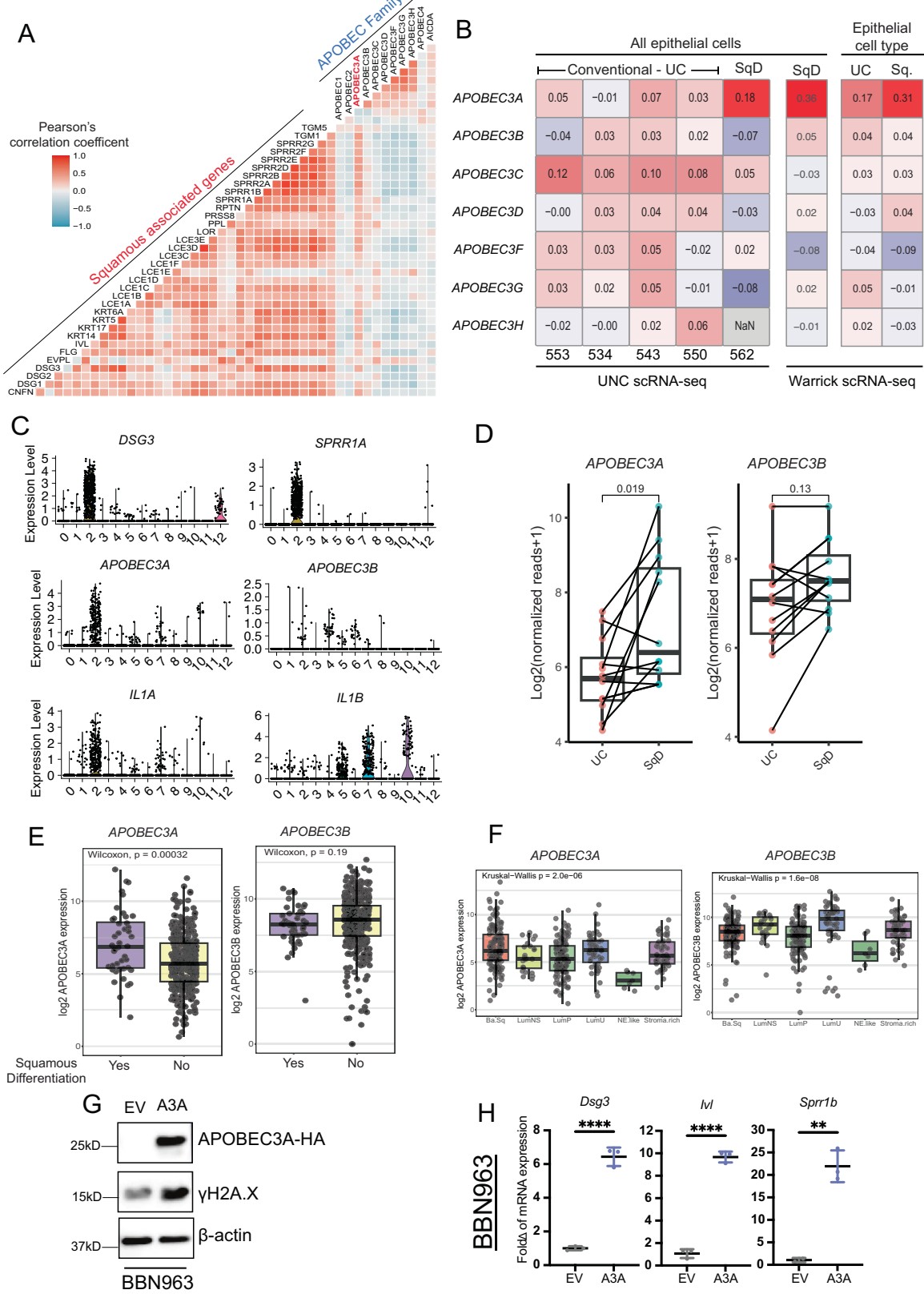

to visualize cleaved and un-cleaved oligonucleotides to quantify the percentage of deaminase activity.

mAPOBEC3 Oligonucleotide: /5MAXN/ATTATTATTATTCCAAT GGATTTATTTATTTATTTATTTATTT

hAPOBEC3A Oligonucleotide: /5MAXN/ATTATTATTATTCTAA TGGATTTATTTATTTATTTATTTATTT

**Quantitative real time PCR**

RNA was extracted using the QIAGEN RNeasy Plus Mini Kit according to the manufacturer's instructions. The cDNA library was synthesized using the iScript™ cDNA Synthesis Kit (Bio-Rad). qPCR was performed using iTaq Universal SYBR Green Supermix (Bio-Rad) on a QuantStudio 6 Flex System according to the

**Fig. 6 | Human A3A, but not A3B, is associated with squamous differentiation.**
**A** Heatmap showing the correlation between the expression of human APOBEC genes and squamous markers in the TCGA dataset. Blue arrow indicates APOBEC3A. **B** Heatmap showing the correlation between expression of human APOBEC3 genes and squamous differentiation signature within the epithelial population from the Warrick et al. and UNC datasets. **C** Cells were grouped by the clustering assignments and violin plots we generated for the indicated genes representing squamous markers, *APOBEC3A* and *APOBEC3B*, and *IL-1α*. **D** Bulk RNA-seq counts for areas of UC and squamous differentiation from Warrick et al. were quantile-normalized and log2-transformed. The pseudo-counts were then plotted for APOBEC3A and APOBEC3B. Two-sided Wilcoxon *p*-values are shown above plot. The center line plotted in the box represents the median, the upper and lower bounds of the box represent the 75th and 25th percentiles, respectively, and the whiskers indicate the minima and maxima values. **E** Box plots of human APOBEC3A and APOBEC3B expression in the TCGA dataset in tumor samples with or without

histologic squamous differentiation. Two-sided Wilcoxon *p*-values are shown above plot. The center line plotted in the box represents the median, the upper and lower bounds of the box represent the 75th and 25th percentiles, respectively, and the whiskers extend 1.5x the interquartile range. **F** Box plots of human APOBEC3A and APOBEC3B expression in the TCGA dataset in each Consensus subtype of bladder cancer. The center line plotted in the box represents the median, the upper and lower bounds of the box represent the 75th and 25th percentiles, respectively, and the whiskers extend 1.5x the interquartile range. **G** Western blots of whole cell lysates from BBN963-EV and BBN963-A3A treated with doxycycline for 7 days, blotted with indicated antibodies. **H** Bar plots showing the expression of squamous markers (*Dsg3*, *Ivl*, and *Sprr1b*) in BBN963-EV and BBN963-A3A cells after 7 days of doxycycline treatment. Significance was calculated using a two-sided unpaired *t*-test. Data represent mean ± SD (*n* = 3, technical replicates), \*\**p* < 0.01, \*\*\*\**p* < 0.0001. Source data are provided as a Source Data file.

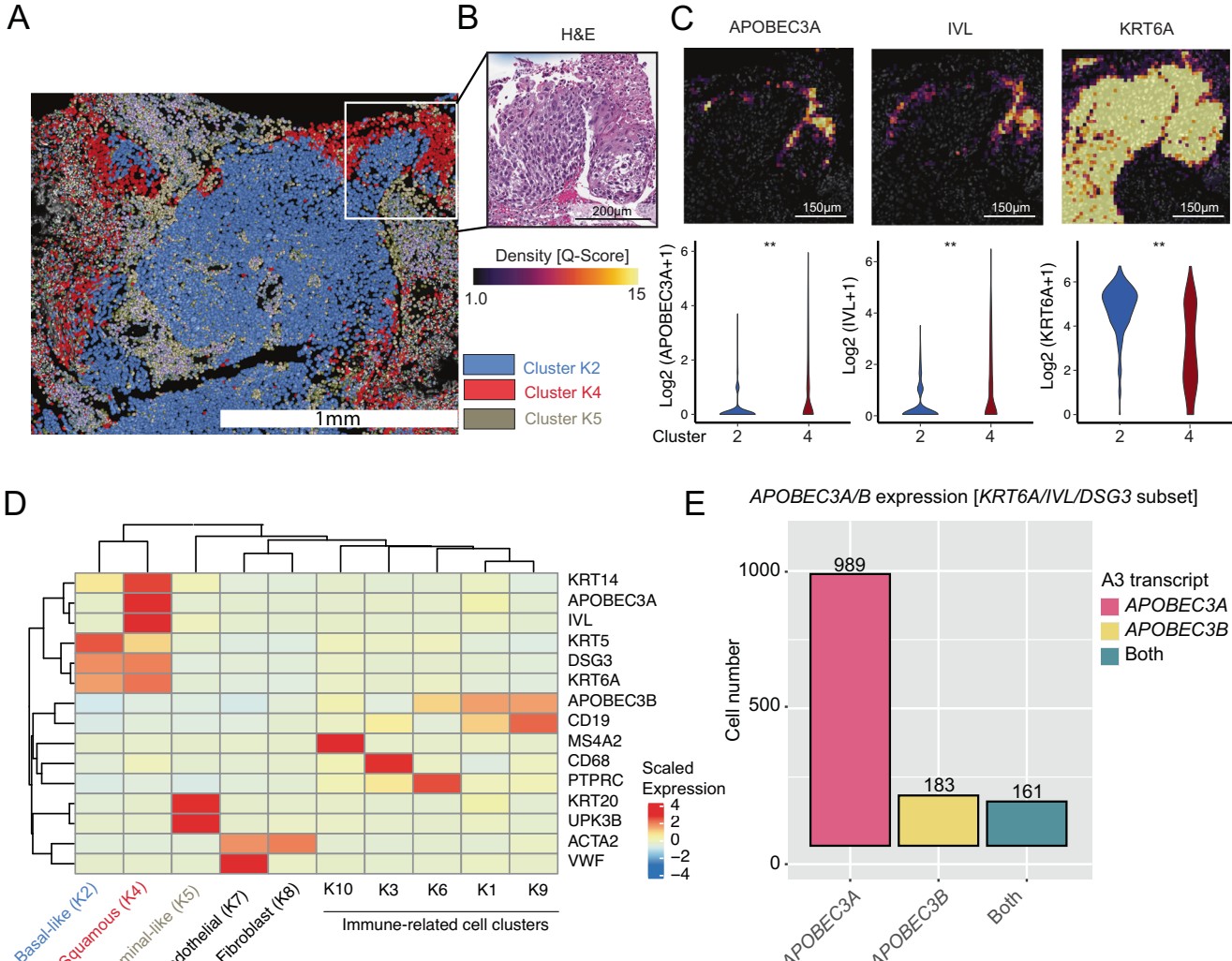

**Fig. 7 | Human APOBEC3A is spatially coexpressed with urothelial carcinoma cells with squamous differentiation.** **A** XeniumRanger processed image of a urothelial carcinoma tumor with 5–10% squamous differentiation. Epithelial cells were pseudo-colored based on XeniumRanger clustering assignment (Cluster K2 = blue, Cluster K4 = red, Cluster K5 = brown). **B** H&E of a region of interest (ROI) containing a squamous cell population aligning with the Cluster K4 cells. **C** Transcript density (top) and transcript quantification by cluster (bottom) for the

indicated gene within the ROI. **D** Heatmap visualizing the relationship between the Cluster assignments and the gene expression for canonical cell markers. Expression is represented by the per-cluster mean transcript count for the indicated gene, calculated for the entire section. **E** Bar graph of the total number of cells (whole section) triple-positive for *KRT6A*, *IVL*, and *DSG3* while also co-expressing *APOBEC3A*, *APOBEC3B*, or both (A3A and A3B). See data availability statement for source data.

manufacturer's instructions. The results were analyzed using the ΔΔCt relative quantification method.

Il1a Forward Primer: CGAAGACTACAGTTCTGCCATT
Il1a Reverse Primer: GACGTTTCAGAGGTTCTCAGAG
Il1b Forward Primer: GCAACTGTTCCTGAACTCAACT
Il1b Reverse Primer: ATCTTTTGGGGTCCGTCAACT
Dsg3 Forward Primer: TGGCAGTCTGGAAGTCACC
Dsg3 Reverse Primer: CTGTAGAGGGTCAGGGATGG
Ivl Forward Primer: AGGAGTCACCTGAGCCAGAACT
Ivl Reverse Primer: TCAGGTGACTCCTGGTACTGCT
Sprr1b Forward Primer: ACCACCTCCTGAGCCATGTGTC
Sprr1b Reverse Primer: AGGGCTCAGGAGCCTTGGGAT
Beta-actin Forward Primer: CATTGCTGACAGGATGCAGAAGG
Beta actin Reverse Primer: TGCTGGAAGGTGGACAGTGAGG
Upk1a Forward Primer: GGCCTGACAGCAAAATAATGA
Upk1a Reverse Primer: GAGAAGCAGGAAGATGGCTT
Upk3a Forward Primer: AGTAGTGCTCAGTGGGACGC
Upk3a Reverse Primer: AGCGGCTCTTACGAGGTTTA

## Immunofluorescence staining

To prepare organoids for paraffin embedment, organoid cultures were washed with 1X PBS and fixed in 2% paraformaldehyde and 0.1% glutaraldehyde for 1 h at room temperature. The culture organoids were gently removed and first embedded in 3% agarose. After agarose solidification, the embedded organoid cultures were incubated overnight at 4 °C in 4% paraformaldehyde. The organoid cultures were transferred and stored in 70% ethanol at 4 °C until paraffin embedding and sectioning. For immunofluorescence labeling, paraffin sections were deparaffinized using xylene and rehydrated through a series of ethanol and 1X PBS washes. Antigen retrieval was performed by boiling slides in a microwave for 10 min in Citrate Buffer (10 mM Citric Acid, 0.05% Tween 20, pH 6.0). Slides were blocked in 5% BSA in PBST for 1 h at room temperature and incubated in primary antibodies diluted in 1% BSA in PBST overnight at 4 °C. The next day, slides were washed with PBST three times, and secondary antibodies were applied for 1 h at room temperature. The slides were sealed with coverslips using VECTASHIELD(R) Antifade Mounting Medium with DAPI (Vector Laboratories).

## ELISA

Cells were plated at a density of 100,000 cells and treated with 4 ug/ml of doxycycline to induce mApobec3 expression. For a 7-day induction, fresh media with doxycycline were replaced every 2–3 days. The supernatant from each cell line was collected, centrifuged at 4 °C at $435 \times g$ for 10 min, and filtered through a 0.45-μm filter to remove cellular debris. A volume of 100 μL of filtered supernatant was added to each well of the Invitrogen mouse IL-1 alpha Uncoated ELISA kit (catalog 88501988). The plate's absorbance was read using a BioTek Gen5 plate reader.

## Single-cell RNA-seq

Single-cell suspensions of tumors were generated using the Miltenyi Biotec Tumor Dissociation Kit and the gentleMACS Dissociator. Red blood cells were lysed using Miltenyi Biotec Red Blood Cell Lysis Solution, and dead cells were removed using the Miltenyi Biotec Dead Cell Removal Kit. Single-cell suspensions were subsequently loaded onto the 10X Genomics Chromium Controller to generate single-cell gel beads in emulsions (GEMs). After GEMs were generated, cDNA libraries were prepared, and libraries were sequenced on an Illumina NovaSeq 6000 in the UNC high-throughput sequencing facility (HTSF).

Using CellRanger, the raw data were processed and mapped to the transcriptome (mm10) to produce a unique molecular identifier (UMI) count matrix. The R Seurat package was used to create a Seurat object using the UMI count matrix. Cells with unique

features, counts >7500, <200, or with >10% mitochondrial counts were filtered out. The data were log- normalized and the most 2500 highly variable features were identified. A linear transformation was then applied, and scRNA-seq datasets for different conditions were integrated using the IntegrateData function in Seurat. Cells are clustered and dimensionally reduced using UMAP with resolution = 0.5. Clusters of distinct cell types were identified and assigned to specific cell types using established markers (ImmGenData) of cell identity (SingleR) on cluster-based methods. Epithelial cells were subset and underwent the same clustering process. Subclusters of epithelial cell biomarkers were identified by the FindAllMarkers function in Seurat. GSEA pathway enrichment test was performed using all significantly differentially expressed genes that overlap with C2, C5, and Hallmark pathways in MSigDB. Trajectory analysis was performed using Monocle (v2.4) on basal, luminal and squamous marker genes.

The Warrick scRNA-seq dataset was downloaded from GEO (GSE172433). The R Seurat package (v4.0.4) was used to create a Seurat object using the UMI count matrix. Cells with unique features, counts <200, >30,000, and >17.5% mitochondrial counts were filtered out. The data were log-normalized, and the most 10,000 highly variable features were identified. A linear transformation was then applied to the scRNA-seq datasets. Cells are clustered and dimensionally reduced using UMAP with resolution = 0.5. Clusters of distinct cell types were identified and assigned to specific cell types using established markers (HumanPrimaryCellAtlasData) of cell identity (SingleR) on cluster-based methods. Cell-level expression was plotted for the indicated genes.

Count-level bulk RNA-seq data from Warrick et al. was obtained from the corresponding author. The data were upper-quartile normalized, and pseudo-counts were generated (normalized counts+1), followed by log2 transformation. The expression for the indicated genes were then plotted using the ggplot2 package in R.

## Xenium spatial transcriptomics

An FFPE UC tumor that has been previously annotated as having 5–10% squamous differentiation was sectioned (10 μm) onto a Xenium slide (Product# 1000460, 10X Genomics, Pleasanton, California) and processed per manufacturer's instructions. Spatial analysis and decoding was performed on the instrument using the Xenium Human Multi-Tissue and Cancer Panel and a 100-gene custom Expression panel (Product# 1000626 and 1000461, 10X Genomics, Pleasanton, California). Raw data was processed using XeniumRanger and all analysis was performed in Xenium Explorer 3 (10X Genomics, Pleasanton, California) and R Studio (2025.05.0, R version 4.5.1).

## Statistics and reproducibility

Western blots and DNA gels show in Figs. 1D, E, 4A, 5A–D, F–H, J, 6G and Supplemental Figs. 3A, B, 4A, 4C, 5A–C, 7A, B are representative images from experiments that were independently repeated at least three times. qPCR data shown in Figs. 3A, C, G, 4B, C, E, G, H, 5E, 6H and Supplemental Figs. 3C, D, 4B, 7C and ELISA data from Fig. 4D are from experiments that were independently repeated at least three times with similar results. Images shown in Figs. 1F, 3B, F, 4F are representative with similar trends seen across tumor and organoid models. No statistical method was used to predetermine sample size. No data were excluded from analysis. The experiments were not randomized. The Investigators were not blinded to allocation during experiments and outcome assessment. GraphPad Prism (version 10) and R (version 4.1.1) were used for statistical analysis.

## Ethical approvals

All animal studies were reviewed and approved by The University of North Carolina at Chapel Hill Institutional Animal Care and Use Committee (IACUC) and conform to the guidelines of the American

Association for Accreditation of Laboratory Animal Care and the US Public Health Service policy on Human Care and Use of Laboratory Animals.

Deidentified patient samples were obtained under IRB exempt protocol #23-1094, as well as IRB protocols #90-0573 and #12–169, which were approved by the University of North Carolina Institutional Review Board. All patients provided written informed consent and were not compensated for their participation. All patient tissue samples were stored and distributed by the University of North Carolina Tissue Procurement Facility per the requirements of the IRB protocol. All tissue samples used for scRNA-sequencing were exhausted during the course of the study. Distribution of any banked tissue used in this study for disease-specific research may be requested by contacting the corresponding authors. All requests are subject to approval by the University of North Carolina Office of Research. As samples were deidentified prior to distribution for research, sex/gender were unknown to the investigators, and therefore not considered in the study design.

## Reporting summary
Further information on research design is available in the Nature Portfolio Reporting Summary linked to this article.

## Data availability
The scRNA-sequencing data generated in this study have been deposited in the Gene Expression Omnibus (GEO) repository under accession number GSE237016, [https://www.ncbi.nlm.nih.gov/geo/query/acc.cgi?acc=GSE237016]. Raw sequencing data and spatial transcriptomic data have been deposited in the dbGaP under accession number phs003405.v1.p1, [https://dbgap.ncbi.nlm.nih.gov/study/phs003405.v1.p1]. Xenium Ranger output files have been depostited in Zenodo under the accession number 17644360, [https://zenodo.org/records/17644360]. Source data are provided with this paper.

## Code availability
No previously unreported or otherwise custom computer code was generated during this study.

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

## Acknowledgements

We thank Dale Cowley and the UNC Animal Models Core facility for the design and generation of the inducible Apobec3 mouse model. We thank the Pathology Services Core (PSC) for expert technical assistance. Both the PSC and the UNC Animal Models Core Facility are supported in part by an NCI Center Core Support Grant (P30CA016086). This work was supported by NIH F31CA247250 (A.S.T.), F31CA281339 (M.S.S.), NIH R01CA241810 and R01CA292625 (W.Y.K.), UNC Integrated Translational Oncology Program (iTOP), T32CA244125 (K.H.G. and W.B.), the University Cancer Research Fund (W.Y.K. and J.S.D.), and the Thomas M. Mohr Fund for Bladder Cancer Research.

## Author contributions

Study conception: A.S.T., J.S.D., and W.Y.K. Experimental design: M.S.S., A.S.T., J.S.D, and W.Y.K. Sample acquisition and clinical annotation: H.J.T., M.A.B., A.B.S., T.L.R., M.I.M., S.E.W., and K.H.G. Data generation: M.S.S., A.S.T., M.Z., E.D.T., W.B., J.R., U.M., I.A.Z., S.E.W., and J.S.D. Data analysis and interpretation: M.S.S., A.S.T., M.Z., E.D.T., J.S.D., and W.Y.K. Manuscript writing: M.S.S., A.S.T., J.S.D., and W.Y.K. Manuscript editing and review: M.S.S., A.S.T., M.Z., E.D.T., W.B., J.R., U.M., I.A.Z., H.J.T., M.A.B., A.B.S., T.L.R., M.I.M., S.E.W., K.H.G., J.S.D., and W.Y.K.

## Competing interests

The authors declare no competing interests.
