## [Peer Review File · Nature Communications]

APOBEC3 promotes squamous differentiation via IL-1A/AP-1 signaling

Corresponding Author: Dr William Kim

Version 0:

Reviewer comments:

Reviewer #1

(Remarks to the Author)

The manuscript by Troung et al reported the role of APOBEC3 in promoting squamous differentiation and metastases in bladder cancer via IL1A/AP-1. The team developed a novel genetically engineered murine model called UPPA with conditional knock out of Pten and Trp53 with mouse Apobec3 (mA3) overexpression, and observed increased basal/squamous differentiation of bladder tumors. The study was meticulously designed to include GEM model, in vitro models, and independent clinical data sets to illustrate the importance of APOBEC3(A) in urothelial squamous differentiation. It is interesting to notice high APOBEC3 may be associated with increased metastasis. Several points below to consider to strengthen the findings:

- Figure 1: What is the frequency of UPPA vs. UPP mouse models in developing squamous differentiation? Is the squamous differentiation sporadic or common (>50%) in both models?
- Fig. 2A: Do the UPPA-squamous population differ from the UPP-squamous cell populations, implying different biological significance? This differential squamous cell type was reflected in Fig. 2F (the squamous population after node 2 in UPPA only).
- It would be helpful to also include UPP vs. UPPA UMAPs reflecting comparable/differential cell clusters.
- Fig. 4B-C: To determine whether IL1a was involved in the secreted conditioned media from BBN963-mA3 cells, IL1a ELISA will be adequate to confirm the presence of this protein.
- Fig. 6E: The single patient scRNA-seq suggested a modest correlation (0.18) between squamous differentiation signature and APOBEC3A gene expression. It would be more convincing, if possible, to identify patients with squamous features based on histology and capture more squamous cells to compute APOBEC correlation.
- Fig. 7: Are the UPPA lung metastasis enriched in squamous phenotype?
- Squamous UC is more aggressive in nature. Does any clinical dataset suggest increased APOBEC3(A) is associated with UC metastasis in general, or the APOBEC3-associated metastasis is limited to a subset of UC that displays squamous phenotype?

Minor:

- There is literature reporting APOBEC3A mutagenesis and metastatic progression in other cancers (Wormann et al, Nat Cancer 2021; PMID: 35121902) which will need to be acknowledged.
- Line 72: 'One' way to understand...

Reviewer #2

(Remarks to the Author)

In this manuscript, Truong et al. demonstrated that mAPOBEC3 (mA3) promotes bladder tumor formation, squamous trans-differentiation, and metastases through IL1a and downstream activation of the AP-1 transcription factor. More importantly, they showed that these phenotypes can be prevented by blocking IL1a signaling with an anti-IL1a antibody, an FDA-approved therapy. The experiments are well executed. However, this manuscript falls short of demonstrating how mA3 promotes tumor formation. It will be critical to perform additional experiments to determine whether mA3 deaminase activity plays a role in these phenotypes. Given the ongoing debate within the APOBEC field regarding whether mA3 functions analogously to its human homologs (A3A and A3B), these experiments are crucial.

Indeed, a prominent concern regarding the data presented in this manuscript revolves around the absence of direct evidence linking mA3 deaminase activity to tumor initiation, thereby ensuring that it is not merely an artifact of the mouse model overexpressing mA3. Furthermore, the manuscript lacks indications demonstrating that mA3 overexpression results in mutations within these tumors. To address these crucial issues, it is imperative to replicate key experiments throughout the manuscript, incorporating a catalytic dead mutant of mA3 (e.g., E73A). The reviewer would like to see experiments demonstrating the impact of the mA3 catalytic dead mutant on survival curve analysis (Fig. 1C), gene expression profiles (Fig. 3), gH2AX induction (Fig. 4A), and activation of IL1R signaling (Fig. 5). In addition, it is important to provide results showing that mA3 induces mutations in bladder tumors but not in the mA3 catalytic dead mutant.

Other comments

- It remains unclear what the extent of mA3 overexpression is in comparison to endogenous mA3 levels with direct comparison using Western blot qPCR. Additionally, it is essential to better establish the relevance of overexpressing mA3. Providing data demonstrating that mouse tumors overexpress mA3 and that this overexpression drives tumor development will be crucial to justify the significance of these experiments, especially in light of existing evidence in human tumors for the overexpression of A3A and A3B.

- Is mA3 activity detected using a deaminase assay in the tumors?

- In lines 416, 417, and 419, please add references to support the claims made in these sentences.

- Is overexpressed mA3 located in the nucleus?

- The Western blot showing the KO IL1R is missing.

Reviewer #3

(Remarks to the Author)

The authors describe an interesting new model of bladder cancer using forced, bladder-confined murine Apobec3 expression in mice with Trp53 and Pten knockout to accelerate carcinogenesis. It is an interesting model system and the findings on IL1a driving squamous change via cFOS are compelling and of potentially broad relevance in other cancer types. APOBEC mutagenesis is the key process in initiating bladder cancer and driving tumour evolution, so research in this area is very important.

Most human bladder cancer is driven by APOBEC mutagenesis, including NMIBC, but most tumours are not squamous. Is the squamous change a temporary consequence of current APOBEC-activity and would cellular plasticity lead to reversion towards luminal biology when the current episode of APOBEC-mutagenesis had ended? I would like to see a little discussion on this point.

The authors select murine Apobec3 for the forced expression system to avoid immune responses but there are implications for how accurately it models human disease. Is the rationale for selecting to over-express Apobec3 to avoid neoantigens sound when the generation of neoantigens is the exact outcome of Apobec3/APOBEC3 activity in the genome?

The limited homology between murine Apobec3 and the human APOBEC3A/APOBEC3B enzymes, thought to do the damage in human cancers, is a concern. Murine Apobec3 is most similar to human APOBEC3B but even here the homology in the protein sequence is only 36.4%. There isn't a mutational signature that I'm aware of yet for murine Apobec3 and so the similarity in the mutational processes remain unknown. For example, human APOBEC3G was recently described to create a strikingly different mutational signature (PMID:36480186) to APOBEC3A and APOBEC3B (and the protein sequence homology between human 3B and 3G is 57.3%). This is a significant limitation to the study that requires discussion.

Figure 2 is reliant on two replicates of scRNAseq and, whilst I know it is an expensive method, my understanding is that the minimum standard for this kind of study is usually n=4 replicates for each condition. An important statistical comparison (that there should be more squamous cells in the Apobec3 expressing tumours; Fig2C) that is central to the hypothesis was not significant, which calls into question how reliable the remaining descriptive analyses would be if subjected to more data. Does the "Basal squamous" cluster represent a later more differentiated squamous state (based on Krt6b and Lce3c) than the "Intermediate squamous"? The two clusters don't seem to share much in terms of the squamous markers selected for Supp Fig1B. Apobec3 expression did not lead to a significant increase in the grouped number of squamous cells and the numerical increase is also small ~1.2-fold (proportion seems to rise from ~0.39 to ~0.47 in Fig 2C). Despite deriving mathematical significance from the numbers of cells analysed, the biological significance of the low fold changes in squamous genes Fig2D is unclear.

It's not clear to me that the pseudotime analysis adds anything to the narrative, particularly because I can't see any increased Apobec3 in the novel UPPA cell population. Is there a better way to illustrate this and apply a statistical test to improve confidence?

The mA3 over-expression in BBN976 does not stimulate a >2-fold change in the mRNA of the selected squamous markers (Dsg3, Ivl, Sprr1b; Suppl Fig 3A), questioning the biological significance. A better control in these models would have been to include a catalytically dead mutant of mA3 alongside the empty vector to account for deaminase-independent biology (PMID:35121902).

I found the human part of this work in Figure 6 to be much weaker because although there is a good weight of data here, it is all correlative in nature. It would have been nice to see the mouse findings validated by forced APOBEC3A/APOBEC3B in human cancer cell lines for example. Particularly if the IL1a/cFOS findings were specific to APOBEC3A. The abstract also over-reaches the data here when it states "human APOBEC3A drive squamous differentiation in UC" because the data provided to support this is purely correlative.

The abstract suggests targeting IL1a as a potential therapeutic approach but is this likely to be clinically relevant? Wouldn't

inhibition make the tumours adopt a more luminal phenotype which could make them harder to treat?

Minor points

-I found the Fig1D high-mag images too dark to see much of the detail.

-Line 72 "On" -> "One"

-Line 129 "on" -> "in"

-Line 263 "SCR" and "sgFos" are undefined at first use here.

-Line 352 "well" -> "wells"

-Line 392 is unlikely to be true at the point of publication because the Harris group has a preprint showing this (bioRxiv <https://doi.org/10.1101/2023.02.24.529970>)

-Line 417 The claim about mitochondrial DNA needs a reference

-There are several points in the text where "APOBEC3" is used but "Apobec3" is meant and this could mislead the reader given there are both human and murine sections to this manuscript. Examples include lines 129, 189, 393, 399, 441.

Reviewer #4

(Remarks to the Author)

In this study, the authors posit that the APOBEC3 family plays a crucial role in urothelial carcinoma (UC). Utilizing a genetically engineered mouse model, they observed that Apobec3 promotes tumor formation, squamous cell differentiation, and metastasis in UC. The mechanism involves the IL1a signaling pathway. As a result, the authors suggest that both mouse Apobec3 and human APOBEC3A may trigger squamous cell differentiation in UC, potentially serving as therapeutic targets. Despite proposing a plausible explanation for the association between transitional cell carcinoma and squamous cell differentiation, in my opinion, this work has significant shortcomings and may not be suitable for publication in Nature Communications.

1. Lines 143-146: The authors indicate that overexpression of APOBEC3 (UPPA) in mice leads to prolonged survival, and the tumors exhibit significantly more and larger squamous cell differentiation compared to the control group (UPP, APOBEC3-/low). Could the overexpression of APOBEC3 affect the size of mouse bladder tumors? Please provide at least 5 sets of replicated data to support this. Additionally, squamous cell differentiation is commonly observed in the progression of transitional cell carcinoma. Is the increased squamous differentiation observed in mouse bladder tumors in Figure 1D due to natural tumor progression rather than APOBEC3 overexpression? This is because the tumors in the control mice, as shown in Figure 1D, appear noticeably smaller than those in UPPA. Could the authors provide a comparative analysis of squamous cell differentiation levels in tumors at the same stage? If this cannot be ruled out, then subsequent research would be meaningless.

2. Line 156: Drawing conclusions based on the results of single-cell sequencing from only two replicated samples is not acceptable. It is recommended that the authors validate the single-cell sequencing results through multi-color fluorescence staining or flow cytometry.

3. Figure 3D: Please provide staining results for squamous genes (proteins) in the same urothelial organoids rather than the expression of a single protein in a specific urothelial organoid.

4. Figure 4B: As far as we know, bladder cancer cells generally do not secrete IFNA and IFNB. I am curious about the detection method used by the authors. Could the authors utilize single-cell sequencing data from tumor samples of UPPA mice and UPP mice to confirm the expression of these cytokines?

5. Line 238: The authors validated the role of IL1A/IL1R but did not rule out the potential interference of IL1B/IL1R. As indicated by the results in Figure 4B, tumor cells can self-secrete IL1B, which acts on IL1R.

6. Figure 5: DNA damage induced by mA3 triggers the production of IL1a, activating IL1R in an autocrine or paracrine manner. This activation promotes p38-ERK signaling and downstream AP-1 transcription factor activation, leading to the expression of squamous genes and cellular differentiation, while suppressing the expression of FOXA1, GATA3, and PPARγ. If possible, could the authors further validate these findings through transcriptome sequencing?

7. Line 354: How does mA3 induce the production of IL1A?

8. Can the authors use UPP-Org-mA3 organoids and UPP-Org-EV to validate the promotion of tumor lung metastasis by mA3?

Version 1:

Reviewer comments:

Reviewer #1

(Remarks to the Author)

The authors have thoughtfully addressed most of the comments in the revised manuscript. However, there are a few pending comments that still require responses:

1. Figure 1. While the new data showed UPPA has a higher number of cases displaying squamous differentiation, however it's not addressed as to what extent the squamous differentiation involves in a tumor. UPPA-squamous differentiation involving 5% vs. 75% of a tumor likely demonstrates differential impact/relevance of mA3 and clinical importance.

2. For the New Fig. 2G-H UPPA model, LSL-Apobec3 was expressed at similar levels in all five cell states including UC cells that showed no squamous differentiation. A discussion on the role of mA3 and squamous differentiation based on this sc-RNAseq data is needed.

3. The notion that mA3 drives increased cellstate 4 and 5 in UPPA models will be strengthened by quantification (% of cells) because the overall number of cells sequenced in UPPA is higher than in UPP models.

Reviewer #2

(Remarks to the Author)

The authors have addressed most of my concerns in depth.

However, the validation of the IL1R knockout cells remains missing. How can the authors be certain that their knockout is correct if they are unable to validate it? While Western blot analysis may be challenging due to the lack of a suitable antibody, alternative approaches, such as sequencing or PCR, could be used to confirm the knockout. Validation of the KO cell line is essential and must be included prior to publication.

The WB data showing both nuclear and cytoplasmic localization of mA3 should be included in supplementary figure, as it demonstrates mA3's ability to localize to the nucleus and cause DNA damage.

Reviewer #3

(Remarks to the Author)

I would like to thank the authors for considering all the comments I raised carefully and doing everything in their power to address them. I hope they agree the manuscript is much improved and I look forward to seeing it published.

Reviewer #4

(Remarks to the Author)

The authors have made substantial progress in addressing the reviewers' concerns, including the addition of new experimental data. These efforts are commendable. However, certain key experiments may remain unaddressed due to technical limitations. In addition, further refinement of the manuscript, particularly in clarifying key term definitions and improving the accuracy of mechanistic and phenotypic descriptions, would enhance the clarity and overall impact of the study.

Version 2:

Reviewer comments:

Reviewer #1

(Remarks to the Author)

The authors have addressed all my comments. I look forward to seeing this work published.

Reviewer #2

(Remarks to the Author)

All my comments have now been addressed appropriately

Reviewer #4

(Remarks to the Author)

This version has basically addressed the issues raised last time.

Reviewer #1 (Remarks to the Author):

The manuscript by Troung et al reported the role of APOBEC3 in promoting squamous differentiation and metastases in bladder cancer via IL1A/AP-1. The team developed a novel genetically engineered murine model called UPPA with conditional knock out of Pten and Trp53 with mouse Apobec3 (mA3) overexpression, and observed increased basal/squamous differentiation of bladder tumors. The study was meticulously designed to include GEM model, in vitro models, and independent clinical data sets to illustrate the importance of APOBEC3(A) in urothelial squamous differentiation. It is interesting to notice high APOBEC3 may be associated with increased metastasis. Several points below to consider to strengthen the findings:

We thank the reviewer for their supportive remarks and constructive comments. We have now done further analysis of our samples and performed additional experiments to hopefully address your concerns.

Figure 1: What is the frequency of UPPA vs. UPP mouse models in developing squamous differentiation? Is the squamous differentiation sporadic or common (>50%) in both models?

We apologize that we did not give a more granular account of the numbers and degree of squamous differentiation in our original submission. We have now included this information in the revised MS (**line 154**). H&E slides were provided to a fellowship trained genitourinary surgical pathologist (S.E.W) and reviewed for evidence of squamous differentiation in a blinded fashion. Tumors arising in the UPPA GEM model were significantly more likely to have squamous differentiation than tumors from the control UPP cohort. We have now included this in the main text of the manuscript (**New Fig. 1G**).

New Fig. 1G

Fig. 2A: Do the UPPA-squamous populations differ from the UPP-squamous cell populations, implying different biological significance? This differential squamous cell type was reflected in Fig. 2F (the squamous population after node 2 in UPPA only).

Thank you for this comment as we had not previously addressed this question. As the reviewer points out in our trajectory analysis, the major difference between UPP and UPPA tumors is the increased number of cell state 4 (yellow) cells in UPPA tumors. The limited number of state 4 cells present in UPP tumors did not allow us to make a direct comparison of differences in gene expression between state 4 in UPP and UPPA cells. In light of not being able to perform a direct comparison, we attempted to address the reviewer's question using 2 alternative approaches.

- First, we wanted to define what typified the novel CellState 4 cells. To this end, within only UPPA epithelial cells, we compared CellState 3 to CellState 4 populations. We found that many of the genes upregulated in CellState 4 compared to CellState 3 were associated with squamous differentiation such as (*Cnfn*, *Sprr3*, *Sprr2d*, *Sprr2i*) consistent with our hypothesis that CellState 4 epithelial cells are a more transcriptional and phenotypically squamous population (**Supplementary Fig. 2E**). GSEA demonstrates a high level of enrichment of our squamous signature (**Supplementary Fig. 2F**). These new figures have been added to the manuscript as **Supplementary Fig. 2E**, **Supplementary Fig. 2F** and described in the manuscript (**line 211**).

New Supplementary Fig. 2E

New Supplementary Fig. 2F

- Second, to directly uncover potential differences between squamous UPP and squamous UPPA cells, we first defined cells with squamous differentiation as cells present in states 3+4 (levels of *lv* only become detectable at state 3, i.e. not detectable in states 1,2) and performed differential gene expression analysis between state 3 + state 4 UPPA and state 3 + state 4 UPP epithelial cells. We found that UPPA cells have decreased levels of extracellular matrix proteins such as collagens (*Col3a1*, *Col1a2*, *Col1a1*), osteopontin (*Spp1*) as well as cell surface proteins involved in modulating cell-cell and cell-matrix interactions Galectin1 (*Lgals1*), *MMP3*, and *SPARC*. Lower levels of type 1 (*Col1a2* and *Col1a1*) and 3 (*Col3a1*) collagens would be predicted to have a less dense and stiff ECM, which might correlate with increased tumor cell motility. This paradigm would be consistent with our findings that mA3 was associated with increased metastatic phenotype that we reported in our original submission. However, at this time we have decided to remove the association between mA3 and metastases from the MS and work towards an independent MS describing those findings. Nonetheless, we wanted to share this analysis with the reviewer and have included the DEG volcano plot as **Reviewer Fig. 1**.

Reviewer Fig 1

It would be helpful to also include UPP vs. UPPA UMAPs reflecting comparable/differential cell clusters.

We have included UMAPs of UPP and UPPA in the revised manuscript as **Supplementary Fig. 2A**. Additionally, while re-evaluating the data to address the reviewer's comment, we noted that grouping the cells by trajectory analysis CellState provided more intuitive and robust populations for analysis (**New Fig 2E**). The new categorization and visualization has been added to the main text of the manuscript as **Fig. 2E**. Grouping by cell states has also allowed us to quantify the cells per state and statistically compare their distribution (**New Fig. 2F**).

New Supplementary Fig. 2A

New Fig. 2E

New Fig. 2F

Fig. 4B-C: To determine whether IL1a was involved in the secreted conditioned media from BBN963-mA3 cells, IL1a ELISA will be adequate to confirm the presence of this protein.

To address the reviewer's comment, as well as comments by the other reviewers, we have now performed ELISA for IL-1A on the conditioned media of a series of isogenic BBN963 cells expressing either empty vector (EV), an enzymatic dead mouse A3 (E73A), or a full length mouse A3 (FL) construct. We found that while full length mA3 produced the highest level of secreted IL-1A, the deaminase-dead mutant, mA3-E73A, also induced IL-1A, but to an attenuated degree (**New Fig 4D**).

To confirm the protein difference in IL-1A was not due to varying expression in the A3 constructs, we performed western blots for mA3. Indeed, both E7A3 and mA3 cells expressed similar levels of protein (**New Fig. 4A**). Additionally, as expected, the E7A3 cells did not possess its deaminase activity (**New Supplementary Fig. 4B**), however interestingly, both the E7A3 and mA3 cells induced gamma

H2A.X equally (**New Fig. 4A**). These findings suggest that the DNA damage induced by mA3 may be, in part, independent of its deaminase function. We have added these findings to the manuscript as main text **Fig. 4A, Fig. 4D** and **Supplementary Fig. 4B**.

New Fig. 4D

New Fig. 4A

New Supplementary Fig. 4B

Fig. 6E: The single patient scRNA-seq suggested a modest correlation (0.18) between squamous differentiation signature and APOBEC3A gene expression. It would be more convincing, if possible, to identify patients with squamous features based on histology and capture more squamous cells to compute APOBEC correlation.

We agree with the reviewer that a correlation of 0.18 is modest, however we should point out that upon histologic review of the tumor, only 5-10% of the epithelial were noted to be squamous, which could partly account for the low correlation.

Ideally, the reviewer is correct, increasing the number of squamous samples would lead to more convincing findings, however that is particularly challenging. Tissue for scRNA-seq is required to be prospectively collected from cases with significant amount of tissue (i.e. incident cases), restricting our samples to incident tumors at TURBT. This means, collection, in general, needs to occur without a *priori* knowledge of the presence or degree of squamous differentiation.

In order to make every effort to address this concern, we leveraged a publicly available dataset that had performed scRNA-seq on a tumor with extensive squamous differentiation (Warrick dataset, PMID: 36323682) In line with our findings, analysis of their data showed a higher correlation between *APOBEC3A* and cells with squamous differentiation (0.36). Notably, the urothelial cells from the UC portion in the Warrick et. al. dataset, also has a modest correlation with *APOBEC3A* expression (0.17). This is most likely secondary to the fact that the tumor has “extensive” squamous differentiation and while phenotypically urothelial, it is composed of cells whose gene expression is in transition to being squamous. We have now included the updated figure in the revised manuscript as **New Fig. 6B**

New Fig. 6B

	All epithelial cells					SqD	Epithelial cell type	
	Conventional - UC				SqD		UC	Sq.
	553	534	543	550				
APOBEC3A	0.05	-0.01	0.07	0.03	0.18	0.36	0.17	0.31
APOBEC3B	-0.04	0.03	0.03	0.02	-0.07	0.05	0.04	0.04
APOBEC3C	0.12	0.06	0.10	0.08	0.05	-0.03	0.03	0.03
APOBEC3D	-0.00	0.03	0.04	0.04	-0.03	0.02	-0.03	0.04
APOBEC3F	0.03	0.03	0.05	-0.02	0.02	-0.08	-0.04	-0.09
APOBEC3G	0.03	0.02	0.05	-0.01	-0.08	0.02	0.05	-0.01
APOBEC3H	-0.02	-0.00	0.02	0.06	NaN	-0.01	0.02	-0.03
	553	534	543	550	562			
	UNC scRNA-seq					Warrick scRNA-seq		

Fig. 7: Are the UPPA lung metastasis enriched in squamous phenotype?

We appreciate the reviewer's interest in this aspect of the original submission, however following revision, the amount of new data (single cell and spatial transcriptomics) needed to address all the reviewers' concerns, severely limited our space. Considering this, we felt that while the metastasis phenotype was interesting, it required further development prior to publication and therefore has now been removed from the paper. We hope to publish this work as a separate manuscript on the relationship between A3 and metastasis.

Squamous UC is more aggressive in nature. Does any clinical dataset suggest increased APOBEC3(A) is associated with UC metastasis in general, or the APOBEC3-associated metastasis is limited to a subset of UC that displays squamous phenotype?

These are great questions that we feel we need to address in more depth than was presented in the first submission. As above, any data on the metastatic phenotype has been removed from the revised MS to focus on APOBEC3's association with squamous differentiation. We hope to publish a more complete and in-depth analysis of how A3 regulates UC metastasis.

Minor:

There is literature reporting APOBEC3A mutagenesis and metastatic progression in other cancers (Wormann et al, Nat Cancer 2021; PMID: 35121902) which will need to be acknowledged.

- We thank the reviewer for pointing us to this publication. We will be incorporating it into our second manuscript on A3 and metastases.

Line 72: 'One' way to understand...

- Thank you, we have corrected this.

Reviewer #2 (Remarks to the Author):

In this manuscript, Truong et al. demonstrated that mAPOBEC3 (mA3) promotes bladder tumor formation, squamous trans-differentiation, and metastases through IL1a and downstream activation of the AP-1 transcription factor. More importantly, they showed that these phenotypes can be prevented by blocking IL1a signaling with an anti-IL1a antibody, an FDA-approved therapy. The experiments are well executed. However, this manuscript falls short of demonstrating how mA3 promotes tumor formation. It will be critical to perform additional experiments to determine whether mA3 deaminase activity plays a role in these phenotypes. Given the ongoing debate within the APOBEC field regarding whether mA3 functions analogously to its human homologs (A3A and A3B), these experiments are crucial.

Indeed, a prominent concern regarding the data presented in this manuscript revolves around the absence of direct evidence linking mA3 deaminase activity to tumor initiation, thereby ensuring that it is not merely an artifact of the mouse model overexpressing mA3.

We would like to thank the reviewer for their time evaluating our manuscript. We apologize if it was not clearly stated in the manuscript, our hypothesis was that mApobec3 and APOBEC3A are promoting tumor progression and squamous differentiation. We did not mean to imply that our data was evidence of APOBEC3 driving tumor initiation. As such, we unfortunately did not examine a cohort of mice that exclusively expressed mA3, which would be the appropriate mouse model to answer that question.

However, to try to understand whether mA3 is promoting tumor initiation we have now re-analyzed our mouse data and calculated the proportion of UPP (32/40) and UPPA (37/40) mice that developed detectible tumors via ultrasound. Based on a Fisher's exact test, there is no difference in the tumor incidence between the groups ($p=0.19$), demonstrating that the Apobec3 allele is unlikely affecting tumor initiation/formation. We have now added the below text to ensure our hypothesis was clearly stated and that we conclude that APOBEC3 does not appear to affect tumor initiation (**line 151**).

“Despite UPPA mice demonstrating a shorter tumor latency (Fig. 1C), UPP and UPPA mice had a similar tumor incidence [UPP = 32/40 and UPPA = 37/40, Fisher's exact = 0.19]. Therefore, Apobec3 appears to accelerate tumor progression but not affect tumor incidence.”

Furthermore, the manuscript lacks indications demonstrating that mA3 overexpression results in mutations within these tumors. To address these crucial issues, it is imperative to replicate key experiments throughout the manuscript, incorporating a catalytic dead mutant of mA3 (e.g., E73A). The reviewer would like to see experiments demonstrating the impact of the mA3 catalytic dead mutant on survival curve analysis (Fig. 1C), gene expression profiles (Fig. 3), γ H2AX induction (Fig. 4A), and activation of IL1R signaling (Fig. 5). In addition, it is important to provide results showing that mA3 induces mutations in bladder tumors but not in the mA3 catalytic dead mutant.

c

We understand the reviewer's concerns, and, in response, generated murine urothelial cell lines that express the catalytic dead version of mA3 (E73A) (**New Fig. 4A**), which lacks deaminase activity (**New Supplementary Fig. 4B**). Specifically, we demonstrate that mA3-E73A does not induce squamous gene expression (**New Fig. 3A**), however, that mA3-E73A is sufficient to induce DNA damage, as assessed by gamma H2A.X (**New Fig. 4A**), independent of its deaminase activity. Additionally, it does promote a minimal induction of IL1A, as assessed by ELISA (**New Fig. 4D**), as well as induction of p-cFOS, indicative of AP1 transcription factor activation (**Supplementary Fig. 5A**) and downregulation of PPARG and GATA3 (**Supplementary Fig. 5C**).

Therefore, while deaminase activity does appear to be necessary for mA3 induction of squamous differentiation, it is not required for mA3 induction of DNA damage or induction of AP-1 signaling. Additionally, the loss of mA3 deaminase activity appears to at least partially abrogate IL-1A production (**New Fig. 4D**), which may explain why E73A does not induce squamous gene expression. We have added the following to the Results section (**line 310**).

*“Interestingly, despite having no deaminase activity (**Supplementary Fig. 4B**), BBN963-E73A showed a comparable level of increase in γ H2A.X compared to BBN963-mA3 cells (**Fig. 4A**). Despite this increase in γ H2A.X, BBN963-E73A cells did not promote squamous gene expression (**Fig 3A**) and only an intermediate amount of IL1A (**Fig. 4D**). The ability of APOBEC3 to induce DNA damage in a deaminase independent manner has been shown by previous groups.^{32–35} These results in aggregate suggest that the DNA damage induced by mA3 is independent of its deaminase function but also suggest that induction of squamous differentiation is more closely linked to deaminase activity and potentially the induction of IL1A.”*

New Fig. 4A

New Supplementary Fig. 4B

New Fig. 3A

New Fig. 4D

New Supplementary Fig. 5A

New Supplementary Fig. 5C

Other comments

It remains unclear what the extent of mA3 overexpression is in comparison to endogenous mA3 levels with direct comparison using Western blot qPCR.

We agree with the reviewer that this is an important control for evaluating the physiologic relevance of the system. To examine the level of mA3 expression from our ROSA26-LSL-mA3 allele (exogenous) compared to endogenous mA3, we compared the protein expression of endogenous mA3 in cell lines derived from UPP tumors (n=3) compared to cell lines derived from UPPA tumors (n=3). We see that all the cell lines (UPP and UPPA) express similar levels of endogenous mA3 (lower band) and that the UPPA derived cell lines express exogenous mA3 levels that are slightly increased, but not dissimilar levels of endogenous mA3 (New Fig. 1D). Therefore, while UPPA mouse tumors and cell lines likely overexpress mA3, it does not appear to be at grossly supraphysiological levels. This has now been added to the manuscript as Fig. 1D.

New Fig. 1D

Additionally, it is essential to better establish the relevance of overexpressing mA3. Providing data demonstrating that mouse tumors overexpress mA3 and that this overexpression drives tumor development will be crucial to justify the significance of these experiments, especially in light of existing evidence in human tumors for the overexpression of A3A and A3B. Is mA3 activity detected using a deaminase assay

We have now performed a deaminase assay to examine the functional deaminase activity found in our UPPA model. We see that three independent cell lines derived from UPPA tumors have elevated deaminase activity compared to cell lines derived from three independent UPP tumors (n=3) (**New Fig. 1E**). This has now been added to the manuscript as **New Fig 1E**.

New Fig. 1E

In lines 416, 417, and 419, please add references to support the claims made in these sentences.

Thank you for pointing out this lack of proper referencing. We have now added the below references.

PMID: 23388464 –A3A localization

PMID: 37474103 – A3A responds to mt-RNA in the cytoplasm

PMID: 21368204 – A3A edits mt-RNA in cytoplasm and A3A has hypermutator activity

Is overexpressed mA3 located in the nucleus?

We have now performed a western blot on our BBN963 cells with doxycycline inducible mA3. We see that mA3 (both E73A deaminase dead mutant and full length) localizes in both the nucleus and cytoplasm in our BBN963 cells (**Reviewer Figure 2**). While we acknowledge that mouse A3 is thought to reside predominantly in the cytoplasm, these conclusions were derived primarily by overexpression of fluorescently tagged proteins. Moreover, mA3 did have perinuclear localization in multiple cell types (PMID: 18509452). At this time we have not included this figure in our revision.

Reviewer Figure 2

The Western blot showing the KO IL1R is missing.

- We apologize but despite multiple attempts we were not able to get a publication quality figure of IL1R CRISPR.

Reviewer #3 (Remarks to the Author):

The authors describe an interesting new model of bladder cancer using forced, bladder-confined murine Apobec3 expression in mice with Trp53 and Pten knockout to accelerate carcinogenesis. It is an interesting model system and the findings on IL1a driving squamous change via cFOS are compelling and of potentially broad relevance in other cancer types. APOBEC mutagenesis is the key process in initiating bladder cancer and driving tumour evolution, so research in this area is very important. Most human bladder cancer is driven by APOBEC mutagenesis, including NMIBC, but most tumours are not squamous. Is the squamous change a temporary consequence of current APOBEC-activity and would cellular plasticity lead to reversion towards luminal biology when the current episode of APOBEC-mutagenesis had ended?

I would like to see a little discussion on this point.

The reviewer is correct to point out that most bladder tumors are not squamous, but rather urothelial carcinoma (UC). Up to 30% of patients have some degree of squamous differentiation present within the majority UC tumor. Our data demonstrates that APOBEC3A is sufficient to drive squamous differentiation within the context of UC. However, we acknowledge that there are most certainly alternative mechanisms, which are also sufficient to drive squamous differentiation.

In addition, just to be clear, we are suggesting that APOBEC3A can drive squamous differentiation within a UC tumor but not that APOBEC is the driver of pure squamous bladder cancer (a histologically separate subtype of bladder cancer). We have added some text in the discussion addressing this question (**line 526**).

“As many bladder cancers have evidence of APOBEC-induced mutagenesis it is interesting that only 30% of urothelial carcinoma have squamous differentiation. There may be multiple mechanisms to account for this including the possibility that APOBEC-mutagenesis (and squamous differentiation) is episodic as seen in cancer cell lines [30849372] as well as that there are multiple mechanisms for UC squamous differentiation. We have shown that IL1A is sufficient to induce squamous differentiation and therefore alternative, APOBEC independent pathways that upregulate IL1A may be sufficient to promote squamous differentiation.”

The authors select murine Apobec3 for the forced expression system to avoid immune responses but there are implications for how accurately it models human disease. Is the rationale for selecting to over-express Apobec3 to avoid neoantigens sound when the generation of neoantigens is the exact outcome of Apobec3/APOBEC3 activity in the genome?

The reviewer is correct that the goal of creating the UPPA model was to make a more faithful murine model of bladder cancer and we acknowledge that there would be advantages to overexpression on one of the human APOBEC3 family members. The main concern about expressing a human version of APOBEC3 is the induction of a strong **allo-antigen**, which would not be physiologic.

To follow up on this point, we compared human and mouse APOBEC3. At the amino acid level, human A3A protein has only 36.04% identity to mouse A3, and Human A3B has only 32.69% identity to mouse A3. Moreover, no identical peptide longer than 5 amino acids is present in either comparison. Therefore, human A3A and A3B would be potent source of **allo-antigens** in a mouse model (akin to the

classic chicken OVA [ovalbumin]) and not physiologic. Ultimately our goal is to make a faithful murine model that could be used for therapeutic studies in the future and wanted to avoid a strong allo-antigen.

Figure 2 is reliant on two replicates of scRNAseq and, whilst I know it is an expensive method, my understanding is that the minimum standard for this kind of study is usually n=4 replicates for each condition. An important statistical comparison (that there should be more squamous cells in the Apobec3 expressing tumours;

We agree that it would be best to have an increased number of samples. Unfortunately, during the revision of the manuscript we had difficulty in maintaining the UPPA colony and have not been able to generate additional GEM tumors. We are therefore limited to the current data (n=2) that we originally used. We apologize for this. Nonetheless, we feel that the functional studies demonstrating that mA3 and A3A are sufficient to induce squamous gene expression and squamous histology, as well as the new human scRNAseq and spatial transcriptomic data, present compelling data that that the hypothesis generating data from our scRNAseq of the UPPA model is valid.

We have now edited the manuscript to emphasize the low replicate number and that the conclusions are hypothesis-generating rather than conclusive.

Fig2C) that is central to the hypothesis was not significant, which calls into question how reliable the remaining descriptive analyses would be if subjected to more data. Does the “Basal squamous” cluster represent a later more differentiated squamous state (based on Krt6b and Lce3c) than the “Intermediate squamous”? The two clusters don’t seem to share much in terms of the squamous markers selected for Supp Fig1B. Apobec3 expression did not lead to a significant increase in the grouped number of squamous cells and the numerical increase is also small ~1.2-fold (proportion seems to rise from ~0.39 to ~0.47 in Fig 2C). Despite deriving mathematical significance from the numbers of cells analysed, the biological significance of the low fold changes in squamous genes Fig2D is unclear.

After reading the reviewers comments and internal discussion, we agree that the cluster-based method of identifying squamous cells (**Original Figure 2A**) was not ideal and did not communicate our observations well. In addition, the list of genes used in the dot/bubble plot (**Original Supplementary Figure 1B**) was based on differential gene expression (using the “find all markers” function in Seurat) and not actually based on a list of squamous genes, which may confuse the reader.

The goal of this analysis was to examine if mA3 expression drives an increased proportion of squamous-like epithelial cells. Evidence to address this question seems most robust from the CellState groups derived in our trajectory analysis (**New Fig. 2E and 2F**). The reviewer can see that there is a clear expansion in CellState 4 (**New Fig. 2E and 2F**). We have used this trajectory based CellState to perform the majority of the analysis in this revisi

New Fig 2E

New Fig 2F

Moreover, we used the unique exogenous sequence of our LSL-mA3 allele to map single cells expressing our exogenous mA3 cDNA. We see that while cells within all 5 cell states express the LSL-mA3 allele, CellStates 4 and 5 express the highest levels (**New Fig. 2G-2H**). We hope that the revised figures better communicate the conclusion that mA3 drives an increased proportion of squamous-like urothelial cells in our novel GEM model.

New Fig. 2G-H

It's not clear to me that the pseudotime analysis adds anything to the narrative, particularly because I can't see any increased Apobec3 in the novel UPPA cell population. Is there a better way to illustrate this and apply a statistical test to improve confidence?

Please see response directly above. We agree that our original pseudotime analysis as presented did not communicate its value. We feel we have now utilized the pseudotime trajectory analysis to better illustrate the conclusion that mA3 drives an increased proportion of squamous-like urothelial cells in our novel GEM model. By assessing expression of the LSL-Apobec3 allele, we also hopefully provide more convincing evidence that the mA3 is driving increased CellState 4 and 5 populations. We certainly agree with the reviewer that in our original submission the value of the pseudotime analysis was not well defined, but now it plays an important role.

The mA3 over-expression in BBN976 does not stimulate a >2-fold change in the mRNA of the selected squamous markers (Dsg3, Ivl, Sprr1b; Suppl Fig 3A), questioning the biological significance.

We agree that the BBN976 cell line data is not as robust as what we see in BBN963 cells. Internally we know that the BBN976 cell line is more basal, whereas the BBN963 are more intermediate basal/luminal. We hypothesized that the less than robust induction of squamous genes is because BBN976 cells are already highly basal/squamous and further induction is minimal. Therefore, we have performed the experiments in an additional, but more luminal, murine bladder cancer cell line, UPFL.1, that was developed after our original submission. In the UPFL.1 cell line, similar to the BBN963 cells, we see robust induction of squamous gene expression (**New Supplementary Fig. 3A**). These results are consistent with our hypothesis that the BBN976 line because of its highly basal/squamous nature, may be difficult to induce further squamous gene expression. We have now included the UPFL.1 data within the supplementary figures (**Supplementary Fig. 3A**).

Additionally, we do have unpublished data that directly compared the relative expression of squamous markers (Dsg3, Ivl, and Sprr1b) between BBN963 and BBN976 cells and we do see that BBN976 cells have significantly higher levels of squamous gene expression at baseline (**Reviewer Figure 3**). Since we did not have data that includes a direct comparison from the UPFL.1 cells we have chosen not to publish this result at this time.

New Supplementary Fig. 3A

Reviewer Fig. 3

A better control in these models would have been to include a catalytically dead mutant of mA3 alongside the empty vector to account for deaminase-independent biology (PMID:35121902).

The suggestion of including a catalytically dead A3, **was also raised by Reviewer 2**. We have provided our prior response below for Reviewer 3's convenience.

We understand the reviewer's concerns, and, in response, generated murine urothelial cell lines that express the catalytic dead version of mA3 (E73A) (**New Fig. 4A**), which lacks deaminase activity (**New Supplementary Fig. 4B**). Specifically, we demonstrate that mA3-E73A does not induce

squamous gene expression (**New Fig. 3A**), however, that mA3-E73A is sufficient to induce DNA damage, as assessed by gamma H2A.X (**New Fig. 4A**), independent of its deaminase activity. Additionally, it does promote a minimal induction of IL1A, as assessed by ELISA (**New Fig. 4D**), as well as induction of p-cFOS, indicative of AP1 transcription factor activation (**Supplementary Fig. 5A**) and downregulation of PPARG and GATA3 (**Supplementary Fig. 5C**).

Therefore, while deaminase activity does appear to be necessary for mA3 induction of squamous differentiation, it is not required for mA3 induction of DNA damage or induction of AP-1 signaling. Additionally, the loss of mA3 deaminase activity appears to at least partially abrogate IL-1A production (**New Fig. 4D**), which may explain why E73A does not induce squamous gene expression. We have added the following to the Results section (**line 310**).

*“Interestingly, despite having no deaminase activity (**Supplementary Fig. 4B**), BBN963-E73A showed a comparable level of increase in γ H2A.X compared to BBN963-mA3 cells (**Fig. 4A**). Despite this increase in γ H2A.X, BBN963-E73A cells did not promote squamous gene expression (**Fig 3A**) and only an intermediate amount of IL1A (**Fig. 4D**). The ability of APOBEC3 to induce DNA damage in a deaminase independent manner has been shown by previous groups.^{32–35} These results in aggregate suggest that the DNA damage induced by mA3 is independent of its deaminase function but also suggest that induction of squamous differentiation is more closely linked to deaminase activity and potentially the induction of IL1A.”*

New Fig. 4A

New Supplementary Fig. 4B

New Fig. 3A

New Fig. 4D

New Supplementary Fig. 5A

New Supplementary Fig. 5C

I found the human part of this work in Figure 6 to be much weaker because although there is a good weight of data here, it is all correlative in nature.

We thank the reviewer for these comments as they are reasonable and fair. Admittedly, **all** human 'omic data are correlative, yet are a key component to high profile papers because they validate that the functional studies are relevant to human tumors. We have reviewed the manuscript and made our best attempt to not imply causality, but rather correlation/association. To bolster our correlative findings we have now included 1) analysis of scRNA-seq from an independent data set (Warrick et. al.) in **New Fig 6B** and **6C**) as well as 2) high-resolution spatial transcriptomics data (10x Genomics Xenium), **New Fig. 7A, 7B, 7C, 7D**).

New Fig. 6B

	All epithelial cells					Luminal cell type		
	Conventional - UC					SqD	UC	Sq.
	553	534	543	550	562			
APOBEC3A	0.05	-0.01	0.07	0.03	0.18	0.38	0.17	0.31
APOBEC3B	-0.04	0.03	0.03	0.02	-0.07	0.05	0.04	0.04
APOBEC3C	0.12	0.06	0.10	0.08	0.05	-0.03	0.03	0.03
APOBEC3D	-0.00	0.03	0.04	0.04	-0.03	0.02	-0.03	0.04
APOBEC3F	0.03	0.03	0.05	-0.02	0.02	-0.08	-0.04	-0.09
APOBEC3G	0.03	0.02	0.05	-0.01	-0.08	0.02	0.05	-0.01
APOBEC3H	-0.02	-0.00	0.02	0.06	NaN	-0.01	0.02	-0.03
	UNC scRNA-seq					Warrick scRNA-seq		

New Fig. 7

It would have been nice to see the mouse findings validated by forced APOBEC3A/APOBEC3B in human cancer cell lines for example. Particularly if the IL1a/cFOS findings were specific to APOBEC3A. The abstract also over-reaches the data here when it states “human APOBEC3A drive squamous differentiation in UC” because the data provided to support this is purely correlative.

We have overexpressed human A3A as suggested by the reviewer (**New Fig. 6G**) and as expected do see that hA3A overexpression is sufficient to drive squamous gene expression (**New Fig. 6H**). This has been included in the main text of the manuscript as **Fig. 6G-H**. We feel that this experiment as well as the additional scRNAseq and spatial transcriptomics data further solidify the link between hA3A and squamous differentiation.

New Fig. 6G

New Fig. 6H

The abstract suggests targeting IL1a as a potential therapeutic approach but is this likely to be clinically relevant? Wouldn't inhibition make the tumours adopt a more luminal phenotype which could make them harder to treat?

The reviewer brings up an interesting point about the correlation between molecular subtype and sensitivity / clinical outcomes to chemotherapy. While in breast cancer it is well established that basal-like tumors have clinical benefit to chemotherapy, in bladder cancer this is much less clear. While early studies (Seiler, Eur Urol, 2017, PMID: 28390739) suggested that akin to breast cancer, only basal bladder tumors had clinical benefit to NAC. Subsequent studies have challenged this notion. Indeed, recent data from the Lund group suggest that luminal tumors of the Uro (urothelial-like) and GU (genomically unstable) subtypes derive similar ORR to cisplatin as the Ba/Sq subtype (PMID: 39348658). Therefore, while the reviewer's comment is an excellent one, we believe that the real-world consequences of IL1A inhibition remain unclear. This is a great question however, and we are committed to testing this in preclinical models in the future. We have added text to the discussion that addresses this interesting point (**line 488**)

“One concern is whether IL-1 blockade could promote phenotypic shift from a basal/squamous tumor to a more luminal UC phenotype thus promoting a tumor that is less responsive to chemotherapy. We previously pointed to the molecular parallels between breast and bladder cancer and suggested that molecular subtype might guide therapy stratification. While in breast cancer it is well established that basal-like tumors (in contrast to luminal) have meaningful clinical benefit from chemotherapy, the findings in bladder cancer are less clear. Early studies suggested that akin to breast cancer, only basal bladder tumors had clinical benefit to NAC. Subsequent studies have challenged this notion. Indeed, recent data from the Lund group suggest that luminal tumors of the Uro (urothelial-like) and GU (genomically unstable) subtypes derive similar response to cisplatin as the Ba/Sq subtype. Therefore, while targeting IL-1A is therapeutically tractable, the ultimate impact remains to be seen.”

Minor points

-I found the Fig1D high-mag images too dark to see much of the detail.

-Line 72 “On” -> “One”
-Line 129 “on” -> “in”
-Line 263 “SCR” and “sgFos” are undefined at first use here.

- We have now addressed all the above minor comments

-Line 352 “well” -> “wells”

- We have chosen to remove the metastasis section, which will be the focus of a future manuscript, therefore this line is no longer included in the manuscript.

-Line 392 is unlikely to be true at the point of publication because the Harris group has a preprint showing this (bioRxiv <https://doi.org/10.1101/2023.02.24.529970>)

- At this time we have chosen not to include the Reuben Harris BioRxiv preprint in our paper since we have removed the metastasis phenotype from this paper. Thank you for pointing us to this citation. We would however point out that their human A3B overexpression is in all cells whereas our model is overexpressing mA3 specifically in tumor cells. Albeit probably a minor point.

-Line 417 The claim about mitochondrial DNA needs a reference

- We have now added the appropriate reference.

-There are several points in the text where “APOBEC3” is used but “Apobec3” is meant and this could mislead the reader given there are both human and murine sections to this manuscript. Examples include lines 129, 189, 393, 399, 441.

- Thank you for these comments. We have gone through the text and when talking about mouse Apobec3 have used lower case. In some instances, we believe that both mouse and human A3 can mediate specific phenotypes and chosen to indicate that as APOBEC3.

Reviewer #4 (Remarks to the Author):

In this study, the authors posit that the APOBEC3 family plays a crucial role in urothelial carcinoma (UC). Utilizing a genetically engineered mouse model, they observed that Apobec3 promotes tumor formation, squamous cell differentiation, and metastasis in UC. The mechanism involves the IL1a signaling pathway. As a result, the authors suggest that both mouse Apobec3 and human APOBEC3A may trigger squamous cell differentiation in UC, potentially serving as therapeutic targets. Despite proposing a plausible explanation for the association between transitional cell carcinoma and squamous cell differentiation, in my opinion, this work has significant shortcomings and may not be suitable for publication in Nature Communications.

1. Lines 143-146: The authors indicate that overexpression of APOBEC3 (UPPA) in mice leads to prolonged survival, and the tumors exhibit significantly more and larger squamous cell differentiation compared to the control group (UPP, APOBEC3-/low). Could the overexpression of APOBEC3 affect the size of mouse bladder tumors? Please provide at least 5 sets of replicated data to support this.

The reviewer is correct to point this out and asks a question we had not previously entertained, whether the size of the bladder tumors in the UPPA mice are different than the control UPP mice. We therefore went back and performed volumetric assessments on the ultrasound images from n=6 UPPA

and UPP mice. We did not see a difference in the size of bladder tumors between UPPA and UPP mice, suggesting that overexpression of Apobec3 does not affect tumor size. We have now included this within our supplementary figures (**New Supplementary Fig.1B**).

New Supplementary Fig. 1B

Additionally, squamous cell differentiation is commonly observed in the progression of transitional cell carcinoma. Is the increased squamous differentiation observed in mouse bladder tumors in Figure 1D due to natural tumor progression rather than APOBEC3 overexpression? This is because the tumors in the control mice, as shown in Figure 1D, appear noticeably smaller than those in UPPA. Could the authors provide a comparative analysis of squamous cell differentiation levels in tumors at the same stage? If this cannot be ruled out, then subsequent research would be meaningless.

We understand the reviewer's concern and as above have ensured that the tumors examined were of the same size (**New Supplementary Fig. 1B**). The tumors pictured in **Fig. 1F** appear to be different in size but that is due to the fact that a significant portion of the UPP tumor was removed to generate scRNA-seq samples as well as organoids. Therefore, the amount of tumor appearing in **Fig. 1F** is only a fraction of what was in the bladder at the time of necropsy.

Moreover, to our knowledge, there is no evidence that squamous differentiation is a consequence of UC cancer progression. Squamous differentiation only happens in up to 30% of UC and while it is associated with higher stage at diagnosis these observations are correlative and more likely reflect that UC with squamous differentiation is more "aggressive" than UC without squamous differentiation not necessarily that larger tumors or higher stage tumors are more likely to have squamous differentiation than smaller or lower stage tumors.

2. Line 156: Drawing conclusions based on the results of single-cell sequencing from only two replicated samples is not acceptable. It is recommended that the authors validate the single-cell sequencing results through multi-color fluorescence staining or flow cytometry.

We agree that it would be best to have an increased number of samples. Unfortunately, during the revision of the manuscript we had difficulty in maintaining the UPPA colony and have not generated additional GEM tumors. We are therefore limited to the current data (n=2) that we originally used. We apologize for this. Nonetheless, we feel that the functional studies demonstrating that mA3 and A3A are sufficient to induce squamous gene expression and squamous histology, as well as the new human scRNAseq and spatial transcriptomic data present compelling data that that the hypothesis generating

data from our scRNAseq of the UPPA model is valid. We have now edited the manuscript to emphasize the low number and that the conclusions are hypothesis-generating rather than conclusive.

3. Figure 3D: Please provide staining results for squamous genes (proteins) in the same urothelial organoids rather than the expression of a single protein in a specific urothelial organoid.

Due to technical reasons, we are not able to examine Ivl, Dsg3, and Krt6a protein expression in the same organoid, the primary antibodies require fluorophore-conjugated secondary antibodies to be visualized. Therefore, it is technically difficult to perform this analysis without staining and destaining the slides. We were able to perform 10x Genomics Xenium spatial transcriptomics on a UC tumor with squamous differentiation to assess co-localization of A3A, A3B, Ivl, Dsg3, and Krt6a as the reviewer requested. This analysis allows us to visualize, in situ, transcript localization at a single cell resolution. We found that of cells expressing A3A, A3B, or both A3A/A3B, A3A expression but not A3B, was more likely in KRT6A/IVL/DSG3 positive cells (**New Fig. 7E**). While not specifically what the reviewer asked for, we hope that this data demonstrates that there is co-expression in the same cell of the RNA that encodes these proteins.

New Fig. 7E

4. Figure 4B: As far as we know, bladder cancer cells generally do not secrete IFNA and IFNB. I am curious about the detection method used by the authors. Could the authors utilize single-cell sequencing data from tumor samples of UPPA mice and UPP mice to confirm the expression of these cytokines?

The reviewer is correct that urothelial cells do not generally express IFNA and IFNB. Upon closer examination at the cT values from the qRT-PCR results, they are quite high, and likely represent minimal to no expression. Therefore, if there is any expression it is more likely due to the primers cross reacting to other transcripts. We have now removed those figure panels from the manuscript and greatly appreciate the reviewer bringing this to our attention.

5. Line 238: The authors validated the role of IL1A/IL1R but did not rule out the potential interference of IL1B/IL1R. As indicated by the results in Figure 4B, tumor cells can self-secrete IL1B, which acts on IL1R.

The reviewer brings up a good point that it is possible that IL1B may be able to drive APOBEC3-induced squamous differentiation. While we know that IL1R signaling is required for the A3-induced squamous differentiation, we had not previously asked whether IL1B was *sufficient* to induce squamous gene expression. We have now treated BBN963 cells with IL-1B and do see that IL-1B is also sufficient to induce squamous gene expression of Dsg3, Ivl, and Sprr1b (**New Supplementary Fig. 4C**).

Interestingly, while the degree of squamous gene induction is equivalent to that seen with IL-1A we do not think that IL-1B plays a major role in A3 induced squamous gene expression for several reasons including (1) the level of mA3 induced IL1A is 10 fold that of IL1B (**Fig 4C**). (2) analysis of scRNAseq from human tumors shows that IL1A, but not IL1B is co-expressed in squamous epithelial cells (**New Fig. 6C**). (3) Similar results are shown in Xenium spatial transcriptomics data (**New Fig. 7D**).

New Supplementary Fig. 4C

New Fig. 6C

New Fig. 7D

6. Figure 5: DNA damage induced by mA3 triggers the production of IL1a, activating IL1R in an autocrine or paracrine manner. This activation promotes p38-ERK signaling and downstream AP-1 transcription factor activation, leading to the expression of squamous genes and cellular differentiation, while suppressing the expression of FOXA1, GATA3, and PPAR γ . If possible, could the authors further validate these findings through transcriptome sequencing?

Thank you for this thoughtful comment. We have now examined bulk RNA-seq data from the Warrick dataset that had sampled paired regions of bladder tumors with both UC and squamous differentiation. We examined gene signatures for PPAR γ , GATA3, and FOXA1 activation and saw that they were all lower in the RNA-seq derived from regions of squamous differentiation. The FOXA1 gene signature was not statistically significant different however. This data has been added as **New Fig. 5F** and text added to **line 343**.

New Fig. 5F

7. Line 354: How does mA3 induce the production of IL1A?

We believe that the promotion of squamous differentiation by mA3 is secondary to its ability to induce DNA damage as evidenced by gamma H2A.X (**New Fig. 4A**), and subsequent production of IL1A. Interestingly however, while deaminase dead version of mA3 (E73A) induced equivalent amounts of gamma H2A.X as full length mA3 (**New Fig. 4A**), secretion of IL1A was significantly lower in E73A expressing cells than full length mA3 (**New Supplementary Fig. 4C**) and did not appear to induce squamous gene expression (**New Fig. 3A**). Therefore, we believe that IL1A (and not necessarily DNA damage specifically) plays a key role in A3-induced squamous differentiation, but there may be threshold of IL1A that needs to be surpassed to induce squamous differentiation. This is backed up by our studies demonstrating that IL1R and downstream IL1R signaling through AP-1 is necessary for mA3-induced squamous gene expression (**Fig. 4F, 4G**).

8. Can the authors use UPP-Org-mA3 organoids and UPP-Org-EV to validate the promotion of tumor lung metastasis by mA3?

We no longer have included the section on the association between mA3 and metastases in our revised MS due to word and figure limits. We are actively developing this story for a second manuscript as well.

REVIEWER COMMENTS

Reviewer #1 (Remarks to the Author):

The authors have thoughtfully addressed most of the comments in the revised manuscript. However, there are a few pending comments that still require responses:

1. Figure 1. While the new data showed UPPA has a higher number of cases displaying squamous differentiation, however it's not addressed as to what extent the squamous differentiation involves in a tumor. UPPA-squamous differentiation involving 5% vs. 75% of a tumor likely demonstrates differential impact/relevance of mA3 and clinical importance.

Thank you for bringing this to our attention. The H&E slides were re-reviewed, and while squamous differentiation was more frequent among UPPA tumors than the UPP. There was no difference in the extent of squamous differentiation, if present. We have now added the following phrase, "When comparing the extent of squamous differentiation between UPP and UPPA tumors, we did not note a difference in the proportion of squamous histology per tumor" (Line 159) and included the below table within the supplementary data (Table S1).

Extent of Squamous Diff.	UPP	UPPA
Extensive (61-99%)	3	6
Moderate (11-60%)	2	4
Minimal (0-10%)	0	1

2. For the New Fig. 2G-H UPPA model, LSL-Apobec3 was expressed at similar levels in all five cell states including UC cells that showed no squamous differentiation. A discussion on the role of mA3 and squamous differentiation based on this sc-RNAseq data is needed.

We have now addressed the role of mA3 and squamous differentiation in the context of our single cell analysis to the discussion (Line 452). For convenience we have reproduced the text below.

"We found that mA3 had profound phenotypic consequences such as promoting squamous differentiation and in the pseudotime analysis from our scRNAseq of UPP and UPPA tumors noted novel cell states of highly squamous cells (cell states 4 and 5) in UPPA tumors. We mapped reads to the LSL-Apobec3 cassette, which allowed us to lineage trace any cells that had undergone Cre-recombination. Interestingly, we found that there were LSL-Apobec3 expressing cells throughout the continuum of cell states suggesting that Apobec3 expression does not immediately promote squamous differentiation. Indeed, this late on-set squamous induction could result from several mechanisms, including the chronic accumulation of Apobec3-induced DNA damage to induce sufficient cytokine production prior to squamous induction or the kinetics of basal/luminal transcription factor downregulation."

3. The notion that mA3 drives increased cellstate 4 and 5 in UPPA models will be strengthened by quantification (% of cells) because the overall number of cells sequenced in UPPA is higher than in UPP models.

This is a good point. We have now plotted the percent of cells expressing mA3 (both endogenous and LSL-mA3) (below figure, top row), as well as the expression of both genes by cell state (below figure, bottom row). In the UPP tumors, where no LSL-Apobec3 is expressed (right), a majority of cells reside in cell states 1-3, whereas in UPPA, while LSL-Apobec3 is expressed across all cell states, the proportion of LSL-mA3 positive cells increase as the cells move from a less differentiated state (cell state 1-2) to a squamous-like state (cell state 4-5). The same trend holds true for endogenous mApobec3. Interestingly, it can be noted that even in the UPP tumors, mA3 expression is elevated along with the proportion of positive cells in cell state 5. We thank the reviewer for their comment and feel that this reinforces our conclusions, as such, we have added this analysis to the manuscript as Supplementary Figure 2D-G.

Reviewer #2 (Remarks to the Author):

The authors have addressed most of my concerns in depth.

However, the validation of the IL1R knockout cells remains missing. How can the authors be certain that their knockout is correct if they are unable to validate it? While Western blot analysis may be challenging due to the lack of a suitable antibody, alternative approaches, such as sequencing or PCR, could be used to confirm the knockout. Validation of the KO cell line is essential and must be included prior to publication.

Thank you for the suggestion. We were able to thaw and culture and perform a western blot using the BBN963-sgEV and BBN963-sgIL1R1 cells from in the conditioned media experiment corresponding to Figure 4H. As shown to the right and now included as Supplementary Figure 4C, the sgIL1R1 reduced the overall abundance of IL-1R1, validating the effective knockdown of our targeted protein.

The WB data showing both nuclear and cytoplasmic localization of mA3 should be included in supplementary figure, as it demonstrates mA3's ability to localize to the nucleus and cause DNA damage.

Thank you for the suggestion. We have now added the western blot to Supplementary Figure 3A.

Reviewer #3 (Remarks to the Author):

I would like to thank the authors for considering all the comments I raised carefully and doing everything in their power to address them. I hope they agree the manuscript is much improved and I look forward to seeing it published.

Thank you very much for your comments and support of our study.

Reviewer #4 (Remarks to the Author):

The authors have made substantial progress in addressing the reviewers' concerns, including the addition of new experimental data. These efforts are commendable. However, certain key experiments may remain unaddressed due to technical limitations. In addition, further refinement of the manuscript, particularly in clarifying key term definitions and improving the accuracy of mechanistic and phenotypic descriptions, would enhance the clarity and overall impact of the study.

We have now reviewed and clarified many our definitions of critical terms and used more precise language regarding the phenotype and mechanisms.

Examples of these updates include:

Clarifying role of Apobec in murine tumor growth in vivo (Line 141-146).

Including an analysis of the extent of squamous differentiation between UPP and UPPA murine tumors. (Line 158)

Additional text regarding Apobec3 expression in the single cell data (also noted by reviewer #1). Line 198 and within the discussion, Line 450.